



# North Atlantic freshwater events influence European weather in subsequent summers

Marilena Oltmanns[1], N. Penny Holliday[1], James Screen[2], D. Gwyn Evans[1], Simon A. Josey[1], Sheldon Bacon[1], and Ben I. Moat[1]

[1]National Oceanography Centre, Southampton, UK
[2]University of Exeter, Exeter, UK

**Correspondence:** Marilena Oltmanns (marilena.oltmanns@noc.ac.uk)

**Abstract.** Amplified Arctic ice loss in recent decades has been linked to increased occurrence of extreme mid-latitude weather. The underlying dynamical mechanisms remain elusive, however. Here, we demonstrate a novel mechanism linking freshwater releases into the North Atlantic with summer weather in Europe. Combining remote sensing, atmospheric reanalyses and model simulations, we show that freshwater events in summer trigger progressively sharper sea surface temperature gradients

in subsequent winters, destabilising the overlying atmosphere and inducing a northward shift in the North Atlantic Current. In turn, the jet stream over the North Atlantic is deflected northward in the following summers, leading to warmer and drier weather over Europe. Our results suggest that growing Arctic freshwater fluxes will increase the risk of heat waves and droughts over the coming decades, and could yield enhanced predictability of European summer weather, months to years in advance.

## 1 Introduction

Arctic near-surface temperature is currently warming twice as fast as the global average (Cohen et al., 2019), which manifests itself in an average sea ice volume loss of $3.0 \pm 0.2 \cdot 1000 \, \text{km}^3 \, \text{decade}^{-1}$ (Kumar et al., 2020). Earlier studies noticed statistical links between the amplified Arctic sea ice loss and an increased occurrence of mid-latitude weather extremes (Francis and Vavrus, 2012; Tang et al., 2014; Screen and Simmonds, 2013; Cohen et al., 2014) but the robustness of these links has been questioned and the underlying mechanisms are poorly understood (Barnes, 2013; Overland et al., 2015; Blackport and Screen,

15  2020).

One potential connection occurs through the ocean. Specifically, the loss of sea ice and glacial ice in the Arctic and sub-Arctic regions constitutes a source of freshwater for the North Atlantic (Carmack, 2000; Bamber et al., 2018), which has been linked to cold surface anomalies and the development of storms in the subpolar region in winter (Oltmanns et al., 2020). Cold anomalies in the subpolar region were, in turn, found to precede heat waves over Europe in the subsequent summer (Duchez

et al., 2016; Mecking et al., 2019). The heat waves were attributed to a stationary jet stream over the North Atlantic (Duchez et al., 2016) and were successfully reproduced in model simulations initialised with the cold anomaly (Mecking et al., 2019). Thus, by triggering cold anomalies in winter, increased surface freshening could initiate a chain of events that first leads to cold anomalies and storms in winter and then heat waves in the subsequent summer.





While earlier studies support individual connections between the North Atlantic sea surface temperature (SST) and the jet
stream (Woollings et al., 2010), or between shifts in the jet stream and heat waves (Dong et al., 2013; Gervais et al., 2020),
the role of freshwater in initiating this causal chain is unclear. Yet, given that the Arctic is expected to continue to warm and
release freshwater from melting ice into the North Atlantic, it is critical to understand how the resulting feedbacks will affect
weather in Europe.

Here, we examine the dynamical mechanisms initiated by freshwater events in the North Atlantic. Combining remote sensing
observations with atmospheric reanalyses, and supported by model simulations, we show that ocean-atmosphere feedbacks
associated with freshwater events exert an influence on European temperatures and precipitation in subsequent summers.

## 2   Data

To estimate the variability of freshwater, we derived two indices that are based on the mean North Atlantic Oscillation (NAO)
in July and August, obtained from the National Oceanic and Atmospheric Administration (NOAA) Climate Prediction Center.
The NAO index was calculated using Rotated Principal Component Analysis, applied to the monthly standardised 500 hPa
geopotential height anomalies between 20° N and 90° N (Barnston and Livezey, 1987).

The analysis of ocean variability includes optimal-interpolated, remote sensing-based SST data from NOAA since 1982
(Reynolds et al., 2002) and gridded, absolute dynamic topography since 1993, distributed by the Copernicus Marine Environ-
ment Monitoring Service (Le Traon et al., 1998). To cover a notable freshwater event in 1980, we extended the NOAA SST
data backward with Hadley Centre HadISST1 data since 1979 (Rayner et al., 2003; Hurrell et al., 2008). However, we did not
go back further in time to focus on the period of increased data quality associated with the onset of satellite observations in
1979.

The ocean data is complemented by output from the ERA5 atmospheric reanalysis model from the European Centre for
Medium-Range Weather Forecasts since 1979 (Hersbach et al., 2018). In addition to the standard variables from ERA5, we
used the maximum Eady growth rate to assess the baroclinic instability in the atmosphere. Following earlier studies (Lindzen
and Farrell, 1980; Dierer et al., 2005), we estimated the maximum Eady growth rate in the 1000 hPa to 750 hPa layer with
$\sigma_E \approx 0.31 \frac{f}{N} \mid \frac{u_{750} - u_{1000}}{z_{750} - z_{1000}} \mid$, where $f$ is the Coriolis frequency, $u$ is the wind speed, $z$ the height, $N$ the Brunt-Väisälä frequency
and the subscripts refer to the associated pressure levels.

To investigate the role of the SST in driving the atmospheric circulation, we employed SST-forced simulations from ECHAM5
(Roeckner et al., 2003) and CAM5 (Neale et al., 2012), obtained from the Facility of Climate Assessments repository (Murray
et al., 2020). ECHAM5 was run with a horizontal resolution of 0.75° x 0.75°, 31 vertical levels and 50 ensemble members,
and CAM5 was run with a horizontal resolution of 1° x 1°, 30 vertical levels and 40 ensemble members. We excluded other
models from the repository that do not cover the period 1979–2018 or that do not include all investigated parameters. To reduce
the influence of increasing greenhouse gas concentrations on the results, we subtracted regionally averaged trends from the air
temperatures, both in ERA5 and the model output.





## 3 Approach

The proposed chain of events begins with the seasonal freshwater release into the North Atlantic each summer. Since an enhanced surface freshening in summer reduces the density, it requires a stronger surface cooling in the subsequent winter to mix the freshwater down. Based on a surface mass balance, the influence of the freshwater on the cooling rate can be used to

infer the new freshening each year from remote sensing-based SST observations (Oltmanns et al., 2020). The inferred, seasonal freshening of the subpolar North Atlantic has been found to be well-correlated with the mean North Atlantic Oscillation (NAO) index in July and August multiplied by -1 (Oltmanns et al., 2020). Therefore, we start with this index as a means to quantify the seasonal surface freshening (Fig. 1).

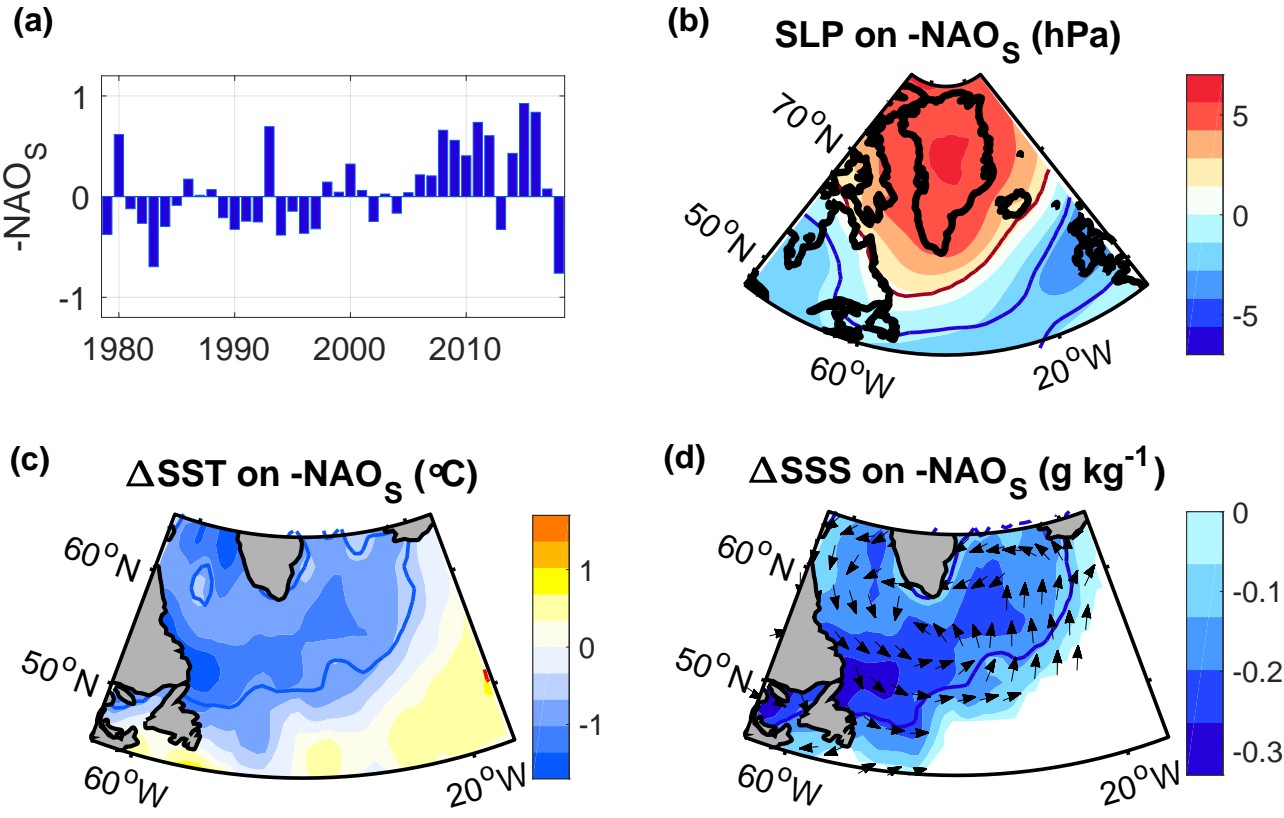

**Figure 1.** (a) Variability of the negative NAO in July and August. (b,c,d) Regression of (b) the sea level pressure in July and August, (c) the SST change and (d) the sea surface salinity (SSS) change from summer (August) to winter (January to March) on the negative summer NAO. The SSS change has been obtained from a surface mass balance (Oltmanns et al., 2020). Thick contours show the 95% confidence levels and arrows indicate the direction of the geostrophic surface flow.





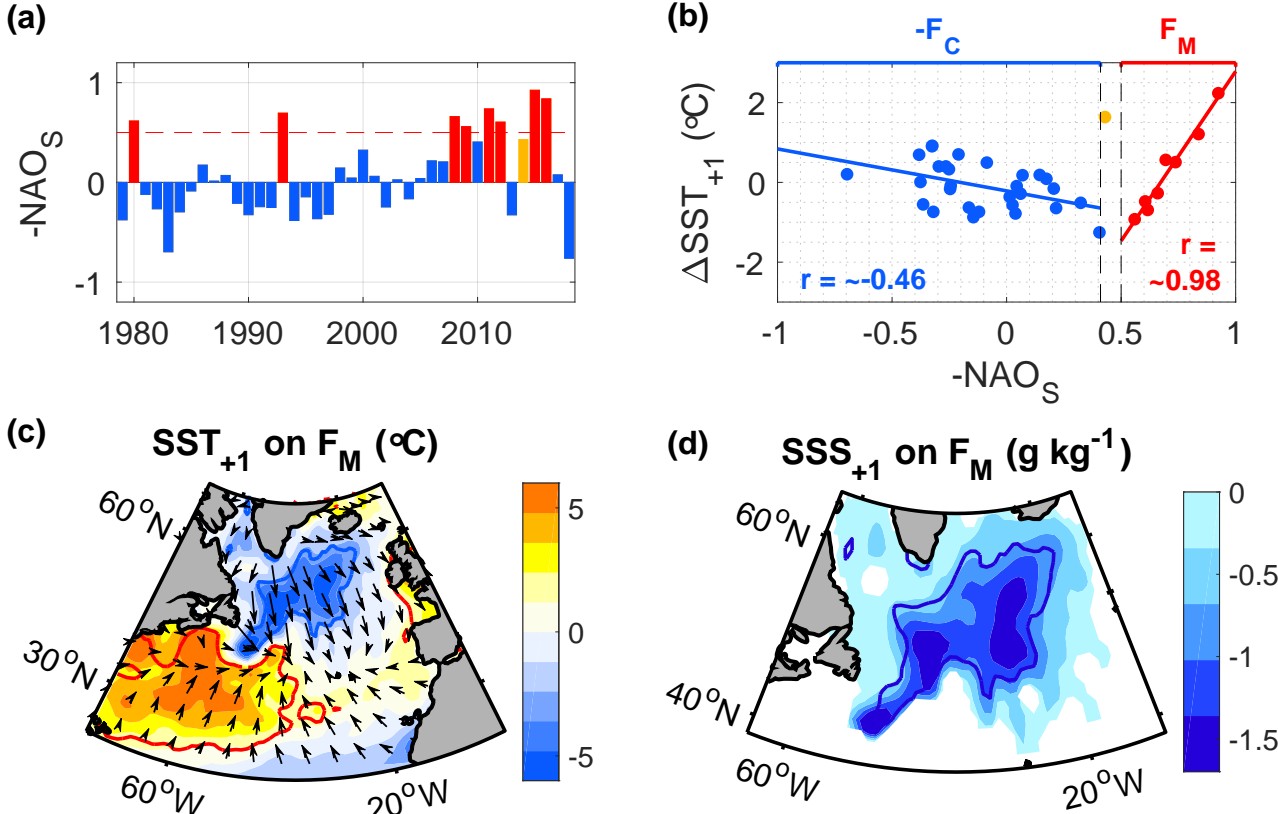

**Figure 2.** (a) Variability of the negative NAO in July and August. Red bars correspond to years, where $-\text{NAO}_S > 0.5$ ($F_M$ years), and blue bars correspond to years, where $-\text{NAO}_S < 0.41$ ($F_C$ years). (b) Correlations of $F_M$ and $-F_C$ with the SST difference between the red and blue 95% confidence region in (c), relative to the climatological mean. (c,d) Regression of (c) the SST and (d) the sea surface salinity, inferred from a surface mass balance (Appendix A), in winter (January through to March) on $F_M$ from the preceding summer. Contours show the 95% confidence levels and the arrows in (c) indicate the direction of the associated, wind-driven Ekman transports.

While the relationship between the negative summer NAO and the seasonal surface freshening is approximately linear, the
response of the North Atlantic SST in the subsequent winter is not. There is a qualitatively different relationship between the negative summer NAO and the SST in the subsequent winter below and above a threshold of $\sim0.5$ in the negative summer NAO. Above this threshold, there is a progressively colder subpolar SST anomaly for higher values of the negative summer NAO. Below this threshold, there is a progressively colder subpolar SST anomaly for smaller values of the negative summer NAO (Fig. 2a-c).

In the following, we will investigate both types of cold anomalies separately. Taking advantage of the near-linear relationships on each side of the threshold, we will use linear regressions to analyse the extent to which both types of cold anomalies are linked to freshwater and examine their influence on European summer weather. Given the overall non-linear distribution





of the SST anomalies with respect to the negative summer NAO, we will assess the significance of the associated links in both directions separately using directional t-tests (Fisher, 1992). For general links, such as the link between the SST and the atmospheric response, we assume near linear relationships over the full observational range and thus assess their significance with non-directional t-tests.

We will further explore the link between fresh anomalies in the subpolar North Atlantic in winter and heat waves over Europe in the subsequent summer by starting from European summer weather. Specifically, we will construct composites of the ocean conditions preceding the warmest and coldest European summers over the last 40 years and assess whether they were significantly different from each other based on two-sample t-tests. Lastly, we will present examples of individual warm summers, sum up the findings, derived from the combined analyses, and discuss potential caveats.

## 4 Results

### 4.1 Melt-induced freshwater events

First, we focus on the cold anomaly that appears above the threshold of $\sim 0.5$ in the negative summer NAO (Fig. 2d). To estimate the associated freshwater anomaly, we use a surface mass balance, taking advantage of the influence of freshwater on the SST. After evaluating the surface fluxes, wind-driven Ekman transports, Ekman pumping and re-emergence of SST anomalies from preceding years, we find that none of these potential drivers are able to explain the observed cold anomaly. They are too weak, not significantly correlated with the index and their distribution is inconsistent with the cooling (Appendix A). Thus, we conclude that the mass increase, implied by the cold anomaly, is balanced by a mass decrease due to a freshwater anomaly. This density compensation of the temperature and salinity anomalies is supported by in-situ observations of recent freshwater events (Oltmanns et al., 2020), and physically means that the amount of surface freshwater sets the temperature that the surface water is required to have before it can be mixed down.

Since the negative summer NAO has previously been linked to increased melting over Greenland (Hanna et al., 2013), we hypothesise that the identified freshwater anomalies result from enhanced runoff, in line with the high pressure anomaly over Greenland (Fig. 1b) and the pronounced seasonality of the associated surface freshening (Fig. 1d). Thus, we refer to these events as melt-induced freshwater events and quantify their strength with the freshwater index $F_M$, which represents the negative summer NAO in the years that it exceeds 0.5 (Fig. 2a and b). We stress, however, that further studies are needed to determine the exact origin of the freshwater. In this study, we will focus on the influences.

The freshwater index $F_M$ has the advantage that its relationship with the cold and fresh anomaly in the subsequent winter is nearly linear, making it a valuable tool to investigate the influences of these freshwater anomalies with linear regressions. While the choice of this index is thus motivated by its strong link to the freshwater anomalies in the subsequent winter, the focus of this study is on the link between freshwater anomalies and their downstream effects, not on the link between the index and the freshwater anomalies. High correlations between the index and the subsequent freshwater anomalies should therefore be regarded as a pre-requisite, not as a conclusion, and we make no assumptions on the usefulness of this index for freshwater anomalies outside the investigated eight years.



## 4.2 Atmosphere-ocean feedbacks in winter

Having identified a set of years with a near-linear relation between the freshwater index and the freshwater anomaly in the subsequent winter, we now exploit this linearity to examine the influences of the freshwater-induced SST anomaly with linear regressions. In accordance with the theory of baroclinic instability (Eady, 1949; Davies and Bishop, 1994), we find that the

sharper SST front between the Gulf Stream and the cold anomaly after stronger relative to weaker freshwater events (Fig. 2c) is associated with an amplified atmospheric instability over a large region around the SST front (Fig. 3a).

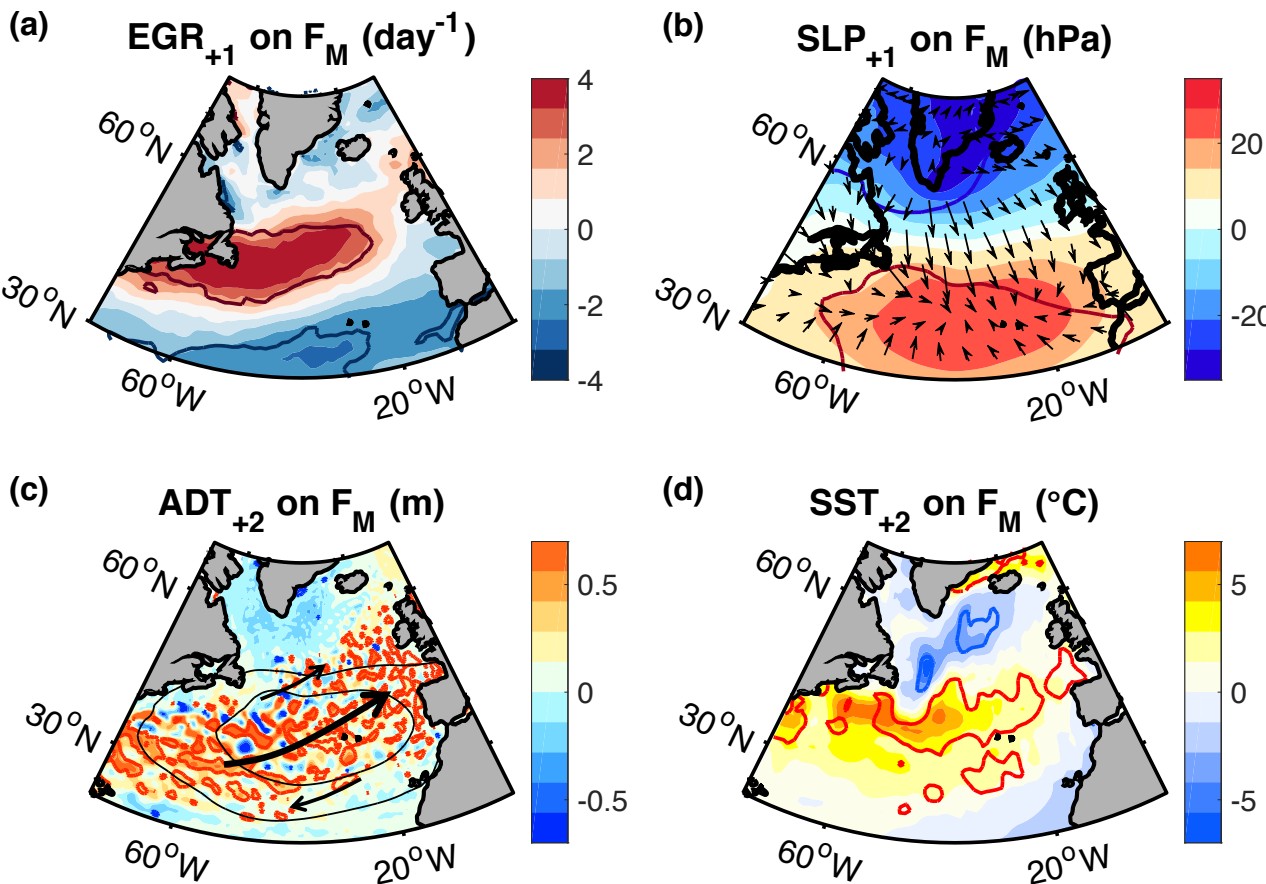

**Figure 3.** Regressions of (a) the maximum Eady growth rate, (b) the sea level pressure, (c) the absolute dynamic topography (ADT) and (d) the SST in winter (January through to March) on $F_M$ from the preceding summer, based on eight winters (Fig. 2d). (a) and (b) are in the first winter after the events whereas (c) and (d) are in the second winter (indicated by the '+1' and '+2' in the title). The arrows in (b) show the direction of the associated Ekman transports, the big arrow in (c) shows the mean location of the North Atlantic Current and the small arrows in (c) indicate the flow implied by the ADT anomaly (see Figure C1 for detailed flow fields). The thin black contours in (c) show the region of Ekman flow convergence, obtained from (b), and the thick contours in all panels delineate the 95% confidence region.





The amplified instability manifests itself in a distinct large-scale atmospheric circulation. When an air parcel travels northward across the anomalous SST front, it rises because it is warmer than the surrounding air masses. By rising, the air column stretches, acquiring positive vorticity. The opposite occurs when an air parcel travels southward across the front. Thus, the SST

front leads to an enhanced baroclinic wave activity, which increases the westerly momentum transfer from the subtropical jet to the eddy-driven polar front jet (O'Reilly et al., 2017; Omrani et al., 2019).

The poleward momentum transfer implies stronger westerlies at mid to high latitudes and weaker westerlies at lower latitudes, which is reflected in a cyclonic anomaly north of the front and an anticyclonic anomaly to the south (Fig. 3b). Thus, the observed atmospheric circulation after stronger compared to weaker freshwater events agrees with the expected, theoretical

response to the underlying SST front. While a detailed investigation of the associated diagnostics is beyond the scope of this study, we confirm that the large-scale dipolar circulation anomaly is reproduced by SST-forced model simulations, supporting that it is driven by the ocean (Appendix B). We point out that the obtained atmospheric circulation anomaly resembles the positive NAO signal and is therefore opposite to that associated with the negative NAO in the preceding summer. This implies that the atmospheric circulation anomaly switches sign, such that the most negative NAO summers are followed by a positive

NAO in the subsequent winter.

The obtained, dipolar atmospheric circulation anomaly drives a convergent Ekman transport between the subtropical and subpolar gyre (Fig. 3b), leading to an increase in sea level height (Fig. 3c). The increased sea level height is manifest in a broad band of anti-cyclonic eddies, whose integrated effect can be described as an inter-gyre gyre circulation (Marshall et al., 2001). The effect of this circulation is a reduced eastward speed at the southern edge of the North Atlantic Current and an increased

eastward speed at the northern edge, implying a northward current shift (Fig. 3c). In the first winter after stronger compared to weaker freshwater events, the northward shift of the North Atlantic Current is obscured by the southward Ekman flow from the cold anomaly, driving enhanced mixing and erosion of the SST front over the eastern North Atlantic (Fig. 2c). However, in the second winter after stronger compared to weaker events, the northward current shift appears as an increasingly sharpened SST front all across to the eastern boundary of the North Atlantic (Fig. 3d).

**4.3 Implications for European summer weather**

The preceding analysis revealed a close link between the magnitude of freshwater events and subsequent winter conditions. Next, we investigate the SST and atmospheric conditions in subsequent summers. In the first summer after freshwater events, we find that the SST is relatively more dominated by the southward expansion of the cold anomaly in stronger compared to weaker events (Fig. 4a). In the second summer, the northward shift of the North Atlantic Current is the most pronounced signal

(Fig. 4b).

As in the preceding winters, the SST fronts destabilise the overlying atmosphere, resulting in an enhanced jet stream along the front (Fig. 4a and b). If the jet stream crosses the downstream coast at a more northerly location, it shields the regions to the south from the moist air over the North Atlantic. Thus, in line with the more northerly SST front and jet stream locations, we observe relatively warmer and drier air over southwest Europe in the first summer after stronger compared to weaker freshwater

events, and relatively warmer and drier air over northwest Europe in the second summer (Fig. 4c-f).



Weather and Climate Dynamics Discussions — Open Access — EGU

**Figure 4.** Regressions of (a,b) the SST with the 700-hPa winds, (c,d) the de-trended 2-m air temperature and (e,f) the accumulated precipitation minus evaporation on $F_M$ in (a,c,e) the first and (b,d,f) the second summer (May through August) after the freshwater events, based on seven summers. We excluded the event in 2015 since its responses were covered by the 2016 event (Fig. C2). The results are not sensitive to excluding all double events (Fig. C3). Thick contours show the 95% confidence levels.



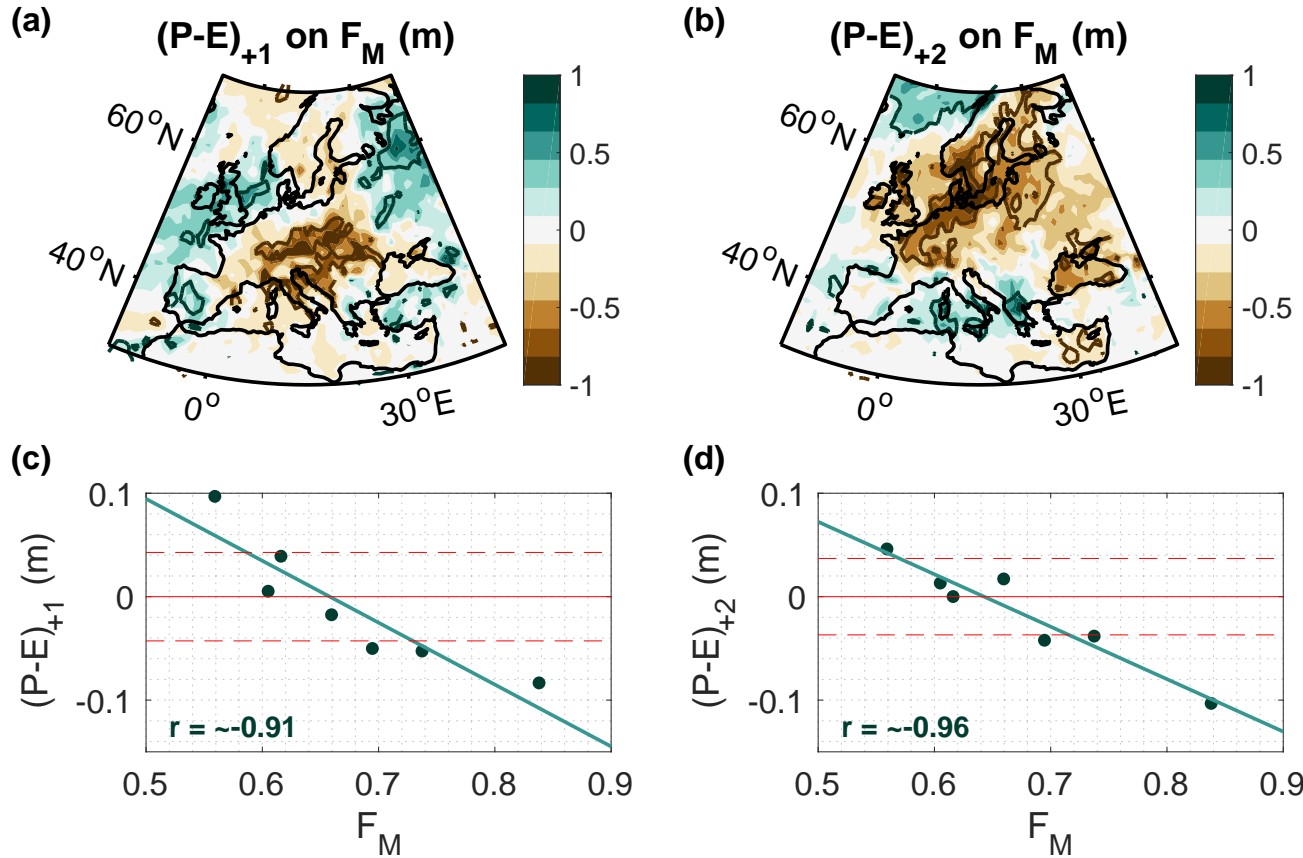

**Figure 5.** Regressions of the accumulated precipitation minus evaporation in (a) the first and (b) the second summer, from May through to August, after freshwater events on $F_M$, again excluding the 2015 event because its responses were covered by the 2016 event (Fig. C2). Contours indicate the 95% confidence levels. (c,d) Correlation between $F_M$ and the accumulated precipitation minus evaporation in the 95% confidence regions in (a) and (b), relative to the climatological summer mean, with dashed lines representing the standard deviation over all summers.

Noting that the regressions of SST and atmospheric circulation on $F_M$ are consistently characterised by steep slopes and high correlations (Figs. 2-4), we conclude that, for the investigated years, where $F_M$ represents the freshwater anomaly extremely well (Fig. 2b), it successfully extracts the downstream effects of the freshwater from other drivers and internal variability. The steep regression slopes further indicate a high sensitivity to small variations in the seasonal freshening. For instance, a

relatively small increase in $F_M$ of $\sim 0.3$, corresponding to an average salinity decrease of $\sim 0.05$ g kg$^{-1}$ over the subpolar region, is associated with a relatively large precipitation decrease of $\sim 15$ cm in the subsequent summer, which is a decrease of more than three standard deviations (Fig. 5c and d).




## 4.4 Simulated atmospheric response to the freshwater-induced SST

To demonstrate the role of the SST pattern in driving the observed atmospheric response, we define an SST index ($SST_{FW}$)
that captures the time variability of the spatial SST pattern linked to freshwater events. Specifically, we map the SST each
summer onto the observed SST pattern after freshwater events. The mapping is obtained from a linear least-square fit of the
SST each summer to the SST pattern obtained from the freshwater events (Fig. 6a and b). Next, we regress the temperature
and precipitation anomalies, obtained from SST-forced model simulations, onto this newly-defined, normalised $SST_{FW}$ index,
using 90 ensemble members from ECHAM5 and CAM5 over 40 years. We find that the observed and simulated atmospheric
responses agree qualitatively well (Fig. 6c-f). While the simulated signals are weaker than the observed ones, and cover a
larger area, the observations lie within the spread of the ensemble members (Fig. 7). Thus, the differences can be reconciled by
internal variability, although we cannot rule out that model biases (Osborne et al., 2020) and the importance of air-sea coupling
contributes to the overall larger magnitudes in the observed regressions.

## 4.5 Circulation-induced freshwater events

Closer inspection of the relationship between the summer NAO and the subsequent SST gradient in winter shows that a positive
summer NAO is also followed by sharper SST gradients (Fig. 2b). Based on a surface mass balance (Appendix A), we again
find that the associated cold anomalies in the subpolar region must be anomalously fresh to account for the density increase
associated with the cooling (Fig. 8).

In contrast to melt-driven freshwater events, the freshwater anomalies associated with a positive summer NAO are preceded
by a more cyclonic atmospheric circulation in the subpolar region, implying a higher wind stress curl and stronger deflection
of freshwater from the Labrador Current into the subpolar gyre (Holliday et al., 2020). While a detailed investigation of the
origin of the freshwater is beyond the scope of this study, the absolute dynamic topography supports the proposed freshening
mechanism (Fig. 9a). In the following, we therefore refer to these freshwater events as circulation-driven events and describe
their strength with $F_C$, corresponding to the summer NAO in the years where it exceeds -0.41 (Fig. 2b).

We sum up that both types of freshwater events are associated with opposite atmospheric circulation anomalies in the
preceding summer. Melt-induced events are associated with more surface freshwater inside the currents while circulation-
induced events result from a change of the currents. However, the surface mass balances do not specify the origin of the
freshwater, nor the location where the balance is attained. For circulation-induced freshwater events in particular, the freshwater
is already anomalously cold when it is advected into the subpolar region, following the geostrophic flow, whereas melt-induced
freshwater events have a strong seasonal signal and the associated cooling occurs locally in the gyre (Fig. 1c).

For circulation-driven freshwater events, the meridional SST-gradient is characterised by higher autocorrelations across the
events, which we attribute to a longer persistence of the associated subpolar gyre circulation. To remove this effect, we exclude
events that are preceded by another strong circulation-driven freshwater event, for which $F_C$ is larger than 0.2. This results in
lower autocorrelations (Fig. A3) and also improves the correlation with the SST gradient from 0.62 to 0.69 (Fig. 8), implying
a better representation of the strength of the freshwater events by $F_C$.



**Figure 6.** (a) Variability and (b) distribution of the SST pattern, linked to freshwater events, obtained by mapping the SST pattern in the box in (b) each summer (May through August) on that after freshwater events. (c,d,e,f) Regressions of (c,e) the observed and (d,f) simulated detrended 2-m air temperature and precipitation minus evaporation in summer onto the $SST_{FW}$ pattern, represented by the normalised $SST_{FW}$ index. The simulations were acquired from 90 SST-forced ensemble members from ECHAM5 and CAM5 over the period 1979–2018. Shown is the mean of the regressions from the ensemble members, not the regression of the mean. Contours indicate the 95% confidence levels.





**Figure 7.** Regressions of (a,b) the SST and (c,d) the accumulated precipitation minus evaporation in summer (May through to August) on the normalised $SST_{FW}$ pattern (Fig. 6a and b), acquired from 50 SST-forced ensemble members from ECHAM5 and 40 ensemble members from CAM5 over the period 1979–2018. Contours show the 95% confidence levels. The box plots show the spread of the ensemble regressions within the central red and brown 95% regions respectively. On each box, the central mark indicates the median, and the left and right edges of the box indicate the 25th and 75th percentiles. The whiskers extend to the most extreme data points excluding outliers, which are plotted using the '+' symbol. The yellow dots represent the corresponding observed regressions.





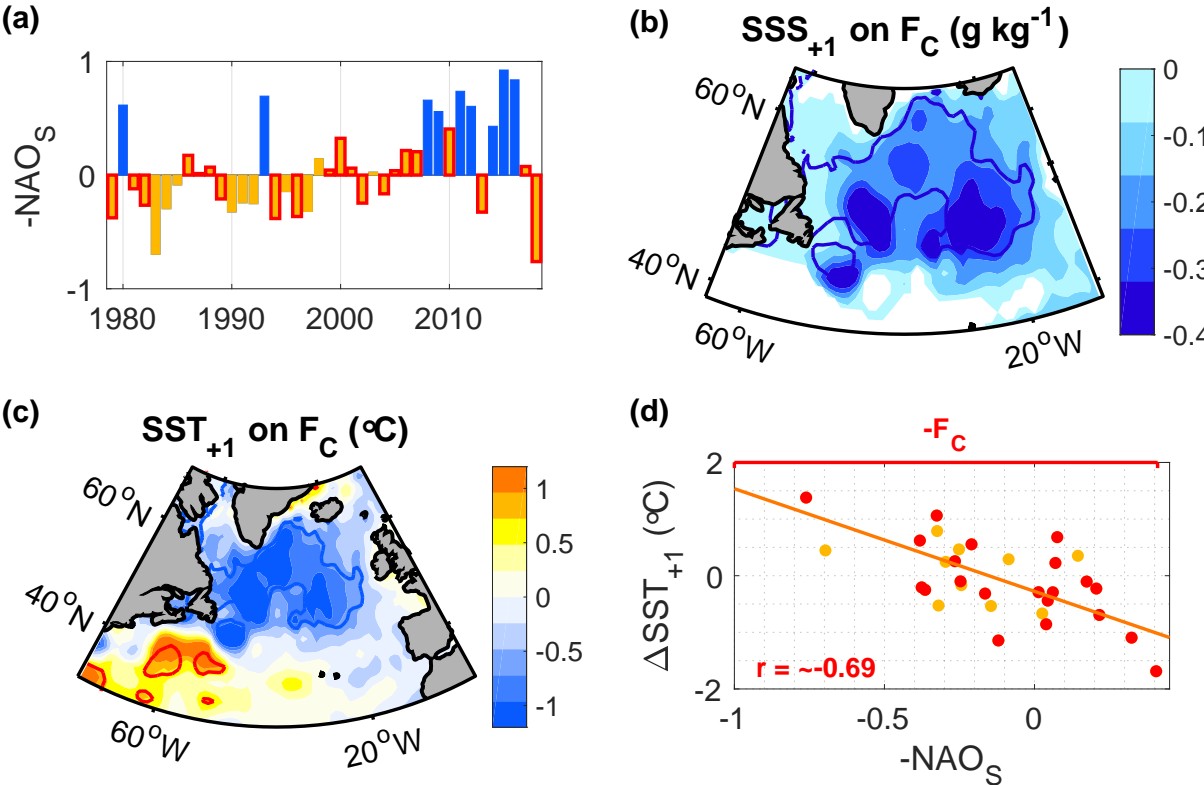

**Figure 8.** (a) Variability of the negative summer NAO with $F_C$ years shown as yellow bars. (b,c) Regression of (b) the surface salinity, inferred from a mass balance, and (c) the SST in winter (January through to March) on $F_C$ from the preceding summer, based on the red years in (a), which remain after dependent events have been excluded (see text for details). Contours show the 95% confidence levels. (d) Correlation between $F_C$ and the SST gradient between the red and blue 95% confidence regions in (c), relative to the climatological mean.

Compared to melt-driven freshwater events, circulation-driven events are followed by a more confined cold anomaly over the central North Atlantic in summer (Fig. 9b). Consequently, the northward deflection of the jet stream occurs further west, leading to more westerly warm and dry anomalies over Europe (Fig. 9c and d). While the regressions for circulation-driven events peak in July and August, and have a smaller extent and amplitude compared those after melt-driven events, the threshold to initiate circulation-driven events is also lower. Thus, we obtained 31 circulation-driven events and 8 melt-driven events between 1979 and 2018 (Fig. 2b).

There was one additional cold anomaly in 2015 that was not captured by either index (Fig. 2a and b). Closer inspection of the atmospheric setting in the preceding summer reveals that it was linked to an unusually large, southward extending anticyclone (Fig. A4a), suggesting the cold anomaly resulted from a melt-driven freshwater event. This is supported by the associated mass balance (Fig. A4b-e) and in-situ freshwater observations in the subpolar North Atlantic (Holliday et al., 2020; Oltmanns et al., 2020). As expected from freshwater events, this event was also followed by a warm European summer in 2015 (Fig. A4f).





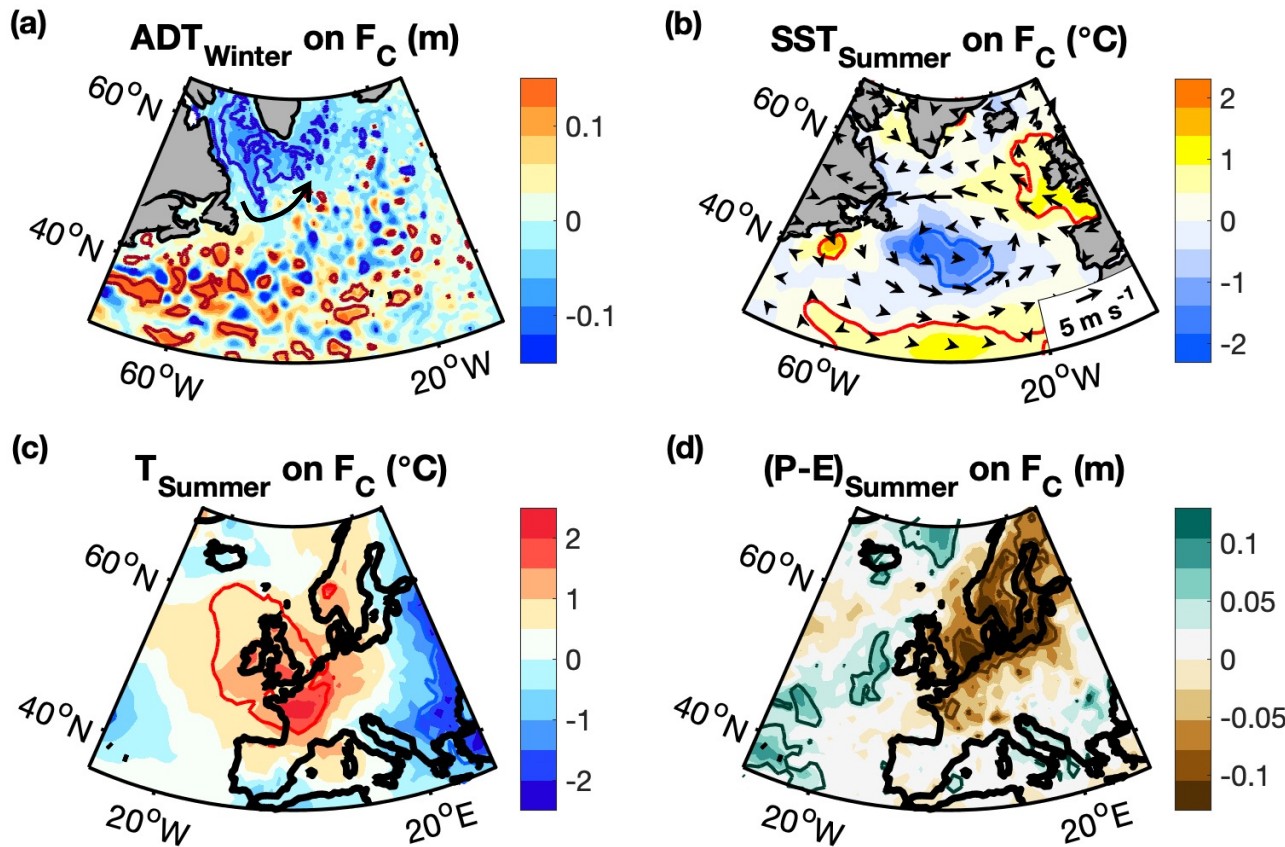

**Figure 9.** Regressions of (a) the absolute dynamic topography (ADT) in winter (January through March), and (b) the SST with the 700-hPa winds, (c) the de-trended 2-m air temperature and (d) the precipitation minus evaporation in summer (July and August) on $F_C$ from the preceding summer, again after removing dependent events, so the regressions are based on 22 events (Fig. 8). The arrow in (a) indicates the direction of the smoothed geostrophic surface flow implied by the ADT anomaly. Contours in all panels show the 95% confidence levels.

## 4.6 Warm summers in Europe

The preceding analyses showed that two types of freshwater events with opposite atmospheric drivers are followed by cold anomalies over the North Atlantic in winter, shifts in the jet stream and warmer, drier weather over Europe in the subsequent summers. Next, we investigate if warm European summers can, in turn, be linked back to a freshwater event in the preceding year.

Based on a composite, we find that the 10 warmest relative to the 10 coldest summers in Europe were associated with a pronounced cold anomaly over the North Atlantic and a northward deflection of the jet stream along the SST front west of Europe, similar to those after freshwater events (Fig. 10a-c). Using a surface mass balance (Appendix A), we trace the cold anomaly back to a freshwater anomaly in the preceding winter (Fig. 10d).





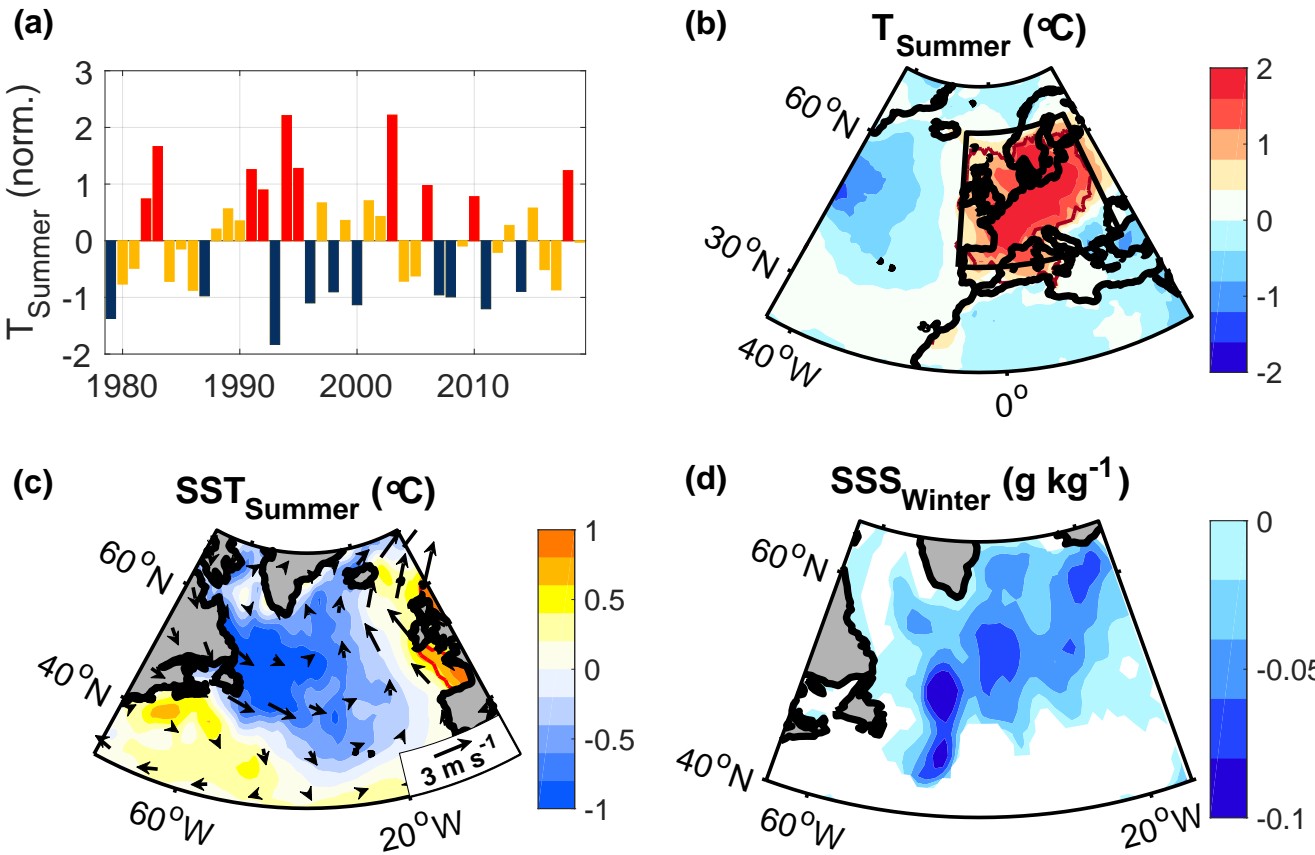

**Figure 10.** (a) Variability of the de-trended 2-m air temperature anomaly in the box shown in (b) during summer (July and August). (b,c) Composites of (b) the de-trended 2-m temperature and (c) the SST and 700 hPa wind anomalies for the ten warmest minus the ten coldest summers, based on (a). Contours indicate the 95% confidence levels. (d) Same as in (b) and (c) but for the sea surface salinity anomaly in the preceding winter, obtained from a surface mass balance.

Since the composite represents an average over many events, it can obscure potential differences between individual events. Thus, we next inspect three warm summers more closely, which occurred in 1994, 2003 and 2018. We selected these summers to demonstrate the role of the location of the North Atlantic cold anomaly for European summer weather but the results are not sensitive to this choice.

The summer in 1994 was characterised by a cold anomaly and jet stream meander that extended far to the south (Fig. 11a and d). In summer 2003, the SST front and jet stream were shifted northward compared to the one in 1994 (Fig. 11b and e). The summer in 2018 was associated with an even further northward SST front and and a particularly large jet stream meander (Fig. 11c and f). As a result, the maximum warm anomaly occurred over southern Europe in 1994, over central Europe in 2003, and over northern Europe in 2018 (Fig. 11g-i).



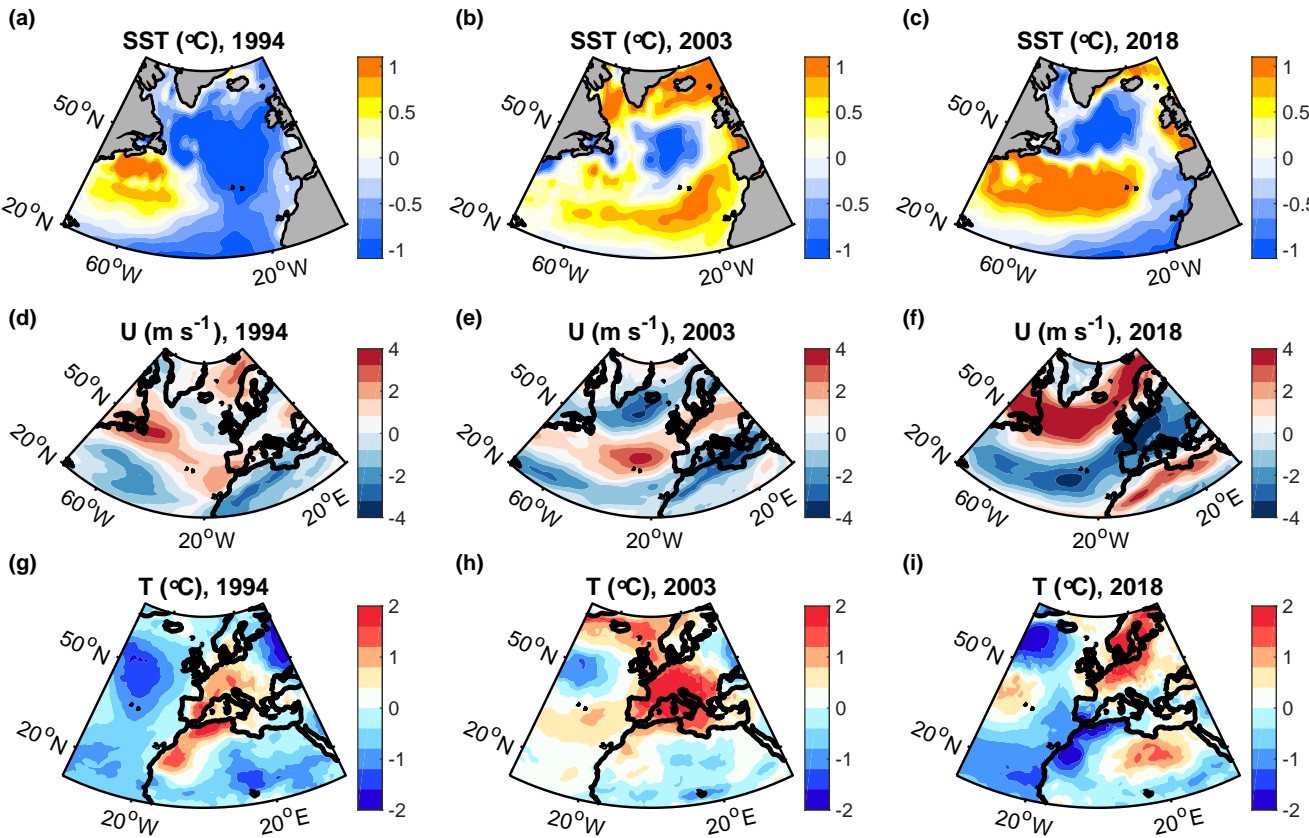

**Figure 11.** Heat wave examples: Anomalies of (a,b,c) the SST, (c,d,e) the eastward wind component at 700 hPa, and (e,f,g) the de-trended 2-m air temperature in selected summers (May through August) to demonstrate how the location of the cold anomaly influences that of the warm anomaly over Europe. Shown are the anomalies relative to the climatological summer means.

According to the preceding analyses of freshwater events, the warm summers in 1994 and 2018 can be linked back to feedbacks associated with the strong melt-induced freshwater events in 1993 and 2016 (Figs. 2-4) while the cold anomaly in summer 2003 was a special case of a circulation-induced freshwater event in 2002 that featured a southward shift of the North Atlantic Current and hence an enhanced southward expansion of cold, fresh polar water (Fig. A6). Thus, all three selected summers support that freshwater played a key role in the development of the cold anomalies and in turn heat waves. Yet,

they also suggest that the exact magnitude, extent and location of the cold anomalies is affected by a combination of different factors, and that further studies are desirable to better understand the evolution of these cold and fresh anomalies in the North Atlantic.





# 5 Conclusions

We surmise that freshwater events can result from enhanced melting and changes in the ocean circulation. Over the recent
period, melt-induced freshwater events have rapidly increased, which we attribute to multi-decadal variability (Kim et al.,
2020) and a potential long-term warming trend. Both types of freshwater events are associated with cold anomalies in the
North Atlantic. In subsequent summers, the jet stream follows the resulting SST front, leading to warmer and drier weather
over Europe.

The relationship between the initial freshwater event and subsequent European summer weather is not always linear and
depends on the type of freshwater event, and the location, strength and extent of the associated North Atlantic cold anomaly. All
of these characteristics are modulated by a combination of ocean and atmospheric feedbacks that remain to be fully understood.
Yet, the existence of dynamical links between North Atlantic freshwater events and European summer weather reveals new
potential to enhance the predictability of European summer weather since current numerical weather prediction systems show
very limited to no forecast skill for European summer weather (Arribas et al., 2011; Dunstone et al., 2018).

Arctic sea ice and glacial ice are expected to further decline in the coming decades (Notz and Stroeve, 2018; Briner et al.,
2020), increasing the freshwater discharge into the North Atlantic. With stronger freshwater events, our results imply an
increased risk of warm, dry European summers and of heat waves and droughts accordingly. Unfortunately, global climate
models have difficulties to capture the hydrographic structure in the subpolar North Atlantic, including the distribution of
freshwater (Menary et al., 2015; Heuzé, 2017; Sgubin et al., 2017; Mecking et al., 2017; Wu et al., 2018). Considering the
identified influences of freshwater on the ocean and atmospheric circulation, our results suggest that models may miss a key
source of climate variability and potential long-range predictability.

*Code and data availability.* This study is only based on publicly available data and standard analysis techniques. The SST and NAO data are
available from NOAA (https://psl.noaa.gov/data/gridded/data.noaa.oisst.v2.html and https://www.cpc.ncep.noaa.gov/products/precip/CWlink/
pna/nao.shtml), absolute dynamic topography data is distributed by the Copernicus Marine Environment Monitoring Service (https://marine.
copernicus.eu/). ERA5 data can be obtained from the European Centre for Medium-Range Weather Forecasts (https://www.ecmwf.int/en/
forecasts/datasets/reanalysis-datasets/era5) and the ECHAM5 and CAM5 model output can be downloaded from the Facility of Climate
Assessments repository (https://psl.noaa.gov/repository/facts). Matlab codes can be obtained from the corresponding author.



**Appendix A: Surface mass balance**

Following earlier studies (Oltmanns et al., 2020), we carried out a scale analysis of the surface mass budget in the cold anomaly
region to estimate the associated freshwater anomaly:

$$\int\limits_{-H}^{0} \frac{\partial \rho}{\partial t} \, dz + \nabla \cdot \int\limits_{-H}^{0} \overrightarrow{u} \rho \, dz = -\frac{B}{g} - M, \tag{A1}$$

where $g$ is the gravitational acceleration, $H$ is an average mixed layer depth of $\sim$250 m, obtained from Argo float climatologies
(Holte et al., 2017), $\overrightarrow{u}$ is the horizontal velocity, $\rho$ is density, $B$ is the downward buoyancy flux through the surface and $M$ is
the downward mass flux through the base of the mixed layer (Griffies and Greatbatch, 2012; Gill, 2016).

After linearising the equation of state $\partial \rho \approx \rho_0 (\alpha \partial T + \beta \partial S)$, where $T$ is the temperature, $S$ is the salinity and $\alpha$ and $\beta$ are
the thermal and haline expansion coefficients, we regressed each term on $F_M$ and $F_C$ (Fig. 2b). There is no contribution from
geostrophic currents to the mass balance as the geostrophic flow is along density contours. Also, both the horizontal Ekman
transport and the vertical Ekman pumping, which represent the largest ageostrophic flow components, are not significantly
correlated with the freshwater indices over the cold anomaly region and their directions are inconsistent with the cold anomaly
(Figs. A1a, b and A2a, b).

Next, we estimated the buoyancy flux anomaly $B = \frac{g\alpha}{c_p} Q + g\beta S (P - E)$, where $c_p$ is the heat capacity, $Q$ is the heat flux
(positive downward) and $P - E$ is the freshwater flux in kg m$^{-2}$ s$^{-1}$ (Gill, 2016). Evaluating the buoyancy flux with the
ERA5 6-hourly surface heat and freshwater fluxes and regressing it on the freshwater indices, we find that it does not match
the distribution of the SST (Figs. A1c and A2c). The surface heat fluxes, which have the largest contribution to the buoyancy
fluxes, are also not significantly correlated with the freshwater indices (Figs. A1d and A2d).

When averaged over the cold anomaly and integrated over the winter, the buoyancy flux after melt-induced freshwater events
reflects an anomalous mass decrease of $\sim$7 kg m$^{-2}$ whereas the cold anomaly implies a mass increase of $\sim$204 kg m$^{-2}$. After
circulation-induced freshwater events, the buoyancy flux reflects an anomalous mass decrease of $\sim$5 kg m$^{-2}$, whereas the cold
anomaly implies a mass increase of $\sim$53 kg m$^{-2}$.

Lastly, we investigated the extent to which reemergence of the cold anomalies from the preceding winters could have con-
tributed to subsequent cold anomalies by investigating the autocorrelations across both types of cold anomalies. For the SST
anomaly associated with $F_M$, we identify no significant autocorrelation even though the corresponding correlation of the SST
with $F_M$ is highly significant (Fig. 2).

For the SST anomaly associated with $F_C$, the autocorrelation is larger, which we attribute to a longer persistence of the
subpolar gyre circulation. To ensure that the cold anomalies associated with $F_C$ are independent, we excluded events that are
preceded by another strong cold anomaly, for which $F_C$ larger than 0.2. This results in lower autocorrelations (Fig. A3) and
also improves the correlation with the SST from $\sim$0.62 to $\sim$0.69. Thus, any potential influence of preceding cold anomalies is
sufficiently small to be negligible (as reflected in the auto-correlations) while, for the same subset of years, the corresponding
correlation of the SST with the $F_C$ remains highly significant, and even increases.



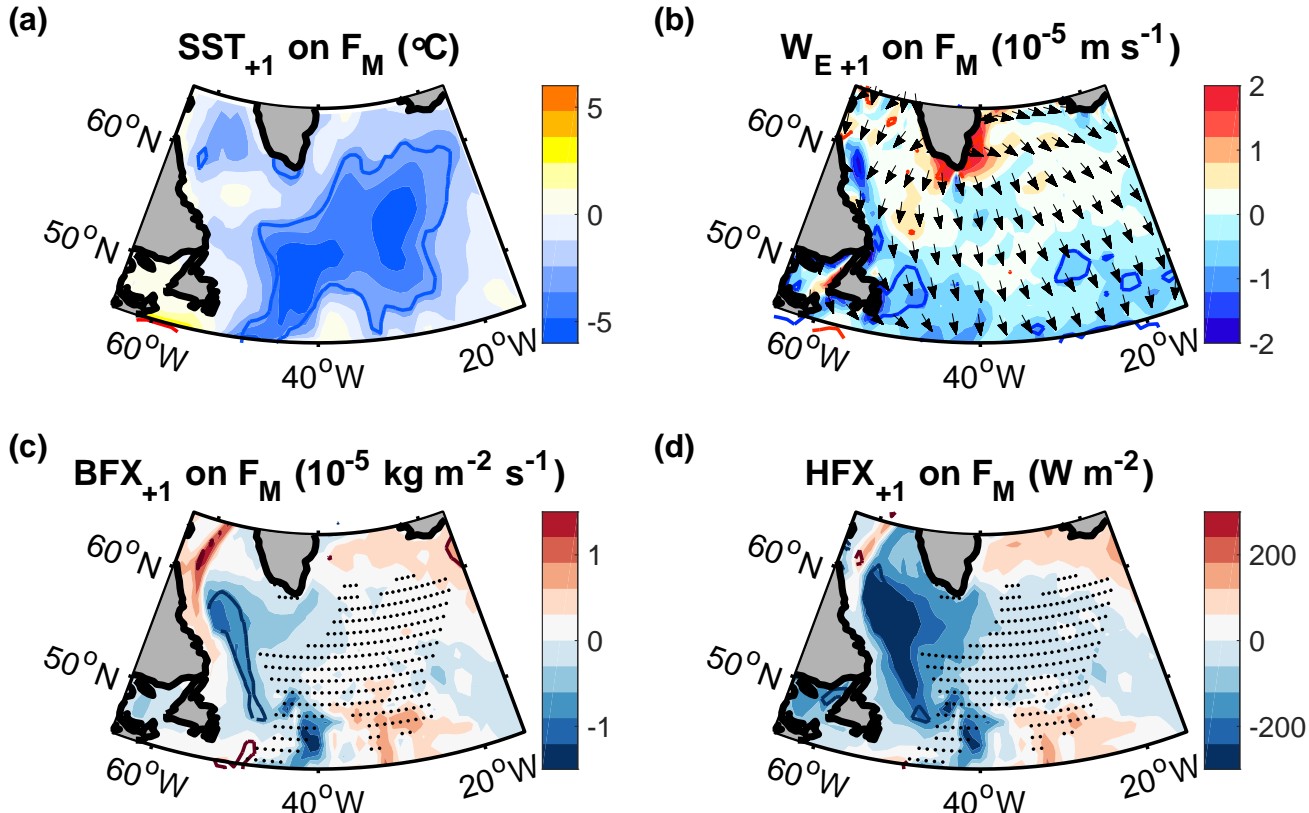

**Figure A1.** Regression of (a) the SST, (b) the vertical Ekman velocity (positive upward), (c) the buoyancy flux anomaly (positive downward) and (d) the surface heat fluxes (also positive downward) in winter (January through March) on $F_M$ from the preceding summer. The arrows in (b) indicate the direction of the horizontal Ekman transports and the dots in (c) and (d) show the region used for the mass balance calculations (corresponding to the cold anomaly region). Contours indicate the 95% confidence levels.

Since subpolar gyre dynamics tend to be characterised by higher autocorrelations, a strong circulation-driven cold anomaly (for which $F_C > 0.2$) could potentially also reinforce a subsequent melt-driven cold anomaly by resulting in a stronger offshore advection of meltwater into the interior gyre. However, none of the investigated melt-driven cold anomalies were preceded by a circulation-induced cold anomaly for which $F_C > 0.2$.

   We conclude that the density increase, reflected by the cold anomaly, is balanced by a density decrease associated with a
freshwater anomaly $(\alpha \Delta T)^* \approx (\beta \Delta S)^*$, where the asterisk indicates the regression on the freshwater indices and $\Delta$ refers to the wintertime anomalies. The balance between the salinity and temperature anomalies has previously been shown to hold during recent freshwater events with in-situ hydrographic observations (Oltmanns et al., 2020). Physically, it means that the freshwater anomaly sets the temperature that the surface water is required to have before it is mixed down.





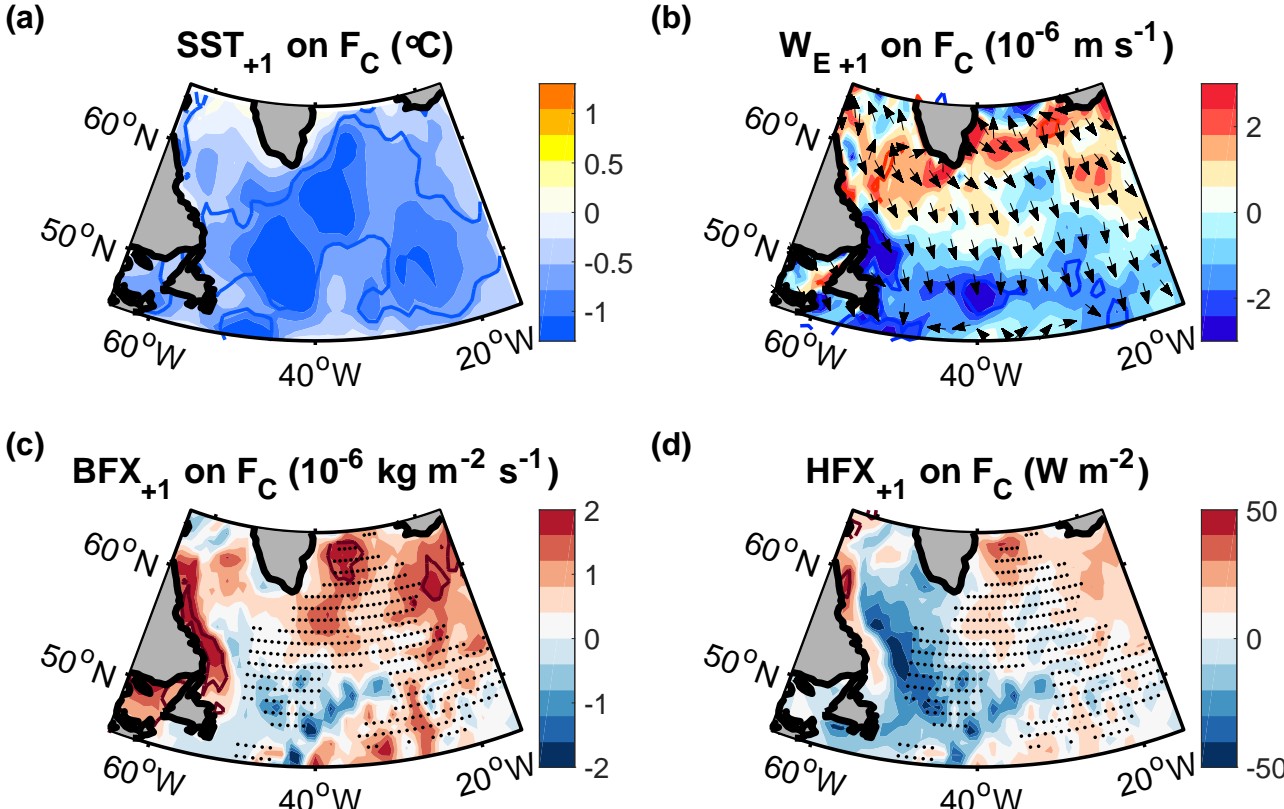

**Figure A2.** Regression of (a) the SST, (b) the vertical Ekman velocity (positive upward), (c) the buoyancy flux anomaly (positive downward) and (d) the surface heat fluxes (also positive downward) in winter (January through March) on $F_C$ from the preceding summer. The arrows in (b) indicate the direction of the horizontal Ekman transports and the dots in (c) and (d) show the region used for the mass balance calculations. Contours indicate the 95% confidence levels.

There was one cold anomaly, which occurred in the winter 2015, that was not covered by either $F_M$ or $F_C$, so we investigated it separately. Using the same mass balance considerations as before, we found that it was associated with a pronounced freshwater event in summer 2014 (Fig. A4). This freshwater event was also confirmed by in-situ hydrographic observations from the subpolar region (Holliday et al., 2020; Oltmanns et al., 2020).

Lastly, we carried out the surface mass balance calculations associated with the composites of the cold anomaly in the winters preceding the 10 warmest relative to the 10 coldest summers over Europe. We obtained qualitatively similar patterns in the potential forcing terms of the cold anomalies compared to those after freshwater events. Again, we find that none of the different drivers, including the surface fluxes, show a significant signal over the cold anomaly region and their amplitudes cannot account for the density increase implied by the cold anomaly (Fig. A5).



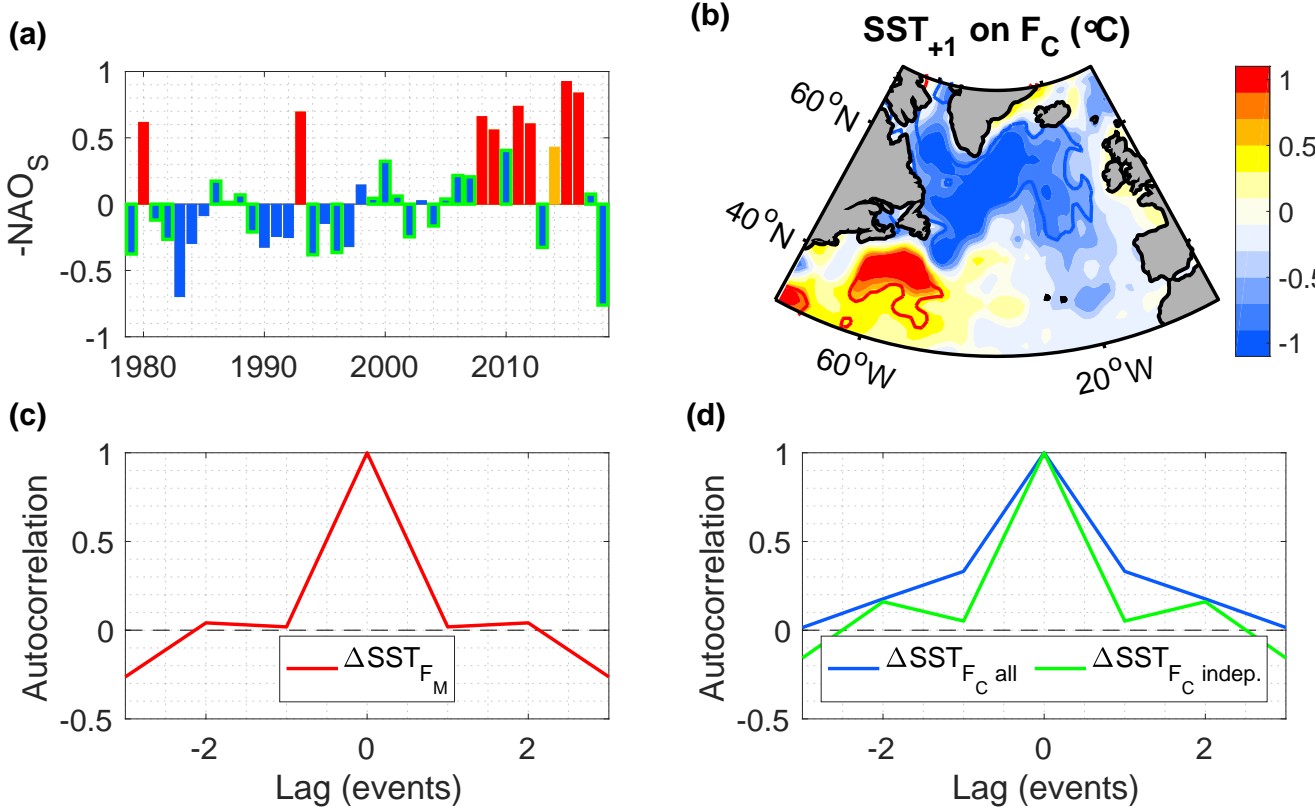

**Figure A3.** (a) Variability of the negative summer NAO in July and August with $F_M$ and $F_C$ years shown as red and blue bars respectively. (b) Regression of the SST in winter (January through March) on $F_C$ from the preceding summer. Contours show the 95% confidence levels. (c,d) Autocorrelations of the SST gradient associated with (c) melt-induced freshwater events, and (d) circulation-induced freshwater events. The SST gradient refers to the difference between the red and blue 95% confidence regions for each type of event (Figs. 2c and 8c). The blue line in (d) represents the autocorrelation across all $F_C$ events, whereas the green line refers to the autocorrelation across independent events (green bars in (a) only). These events remain after removing the events that were preceded by a strong event, for which $F_C > 0.2$.



**Figure A4.** Forcing and influences of the 2014 freshwater event. Anomaly of (a) the SLP in summer 2014, and (b) the SST, (c) the vertical Ekman velocity (positive upward), (d) the buoyancy flux anomaly (positive downward) and (e) the sea surface salinity anomaly in winter 2015, and (f) the de-trended 2-m temperature anomaly in summer 2015.



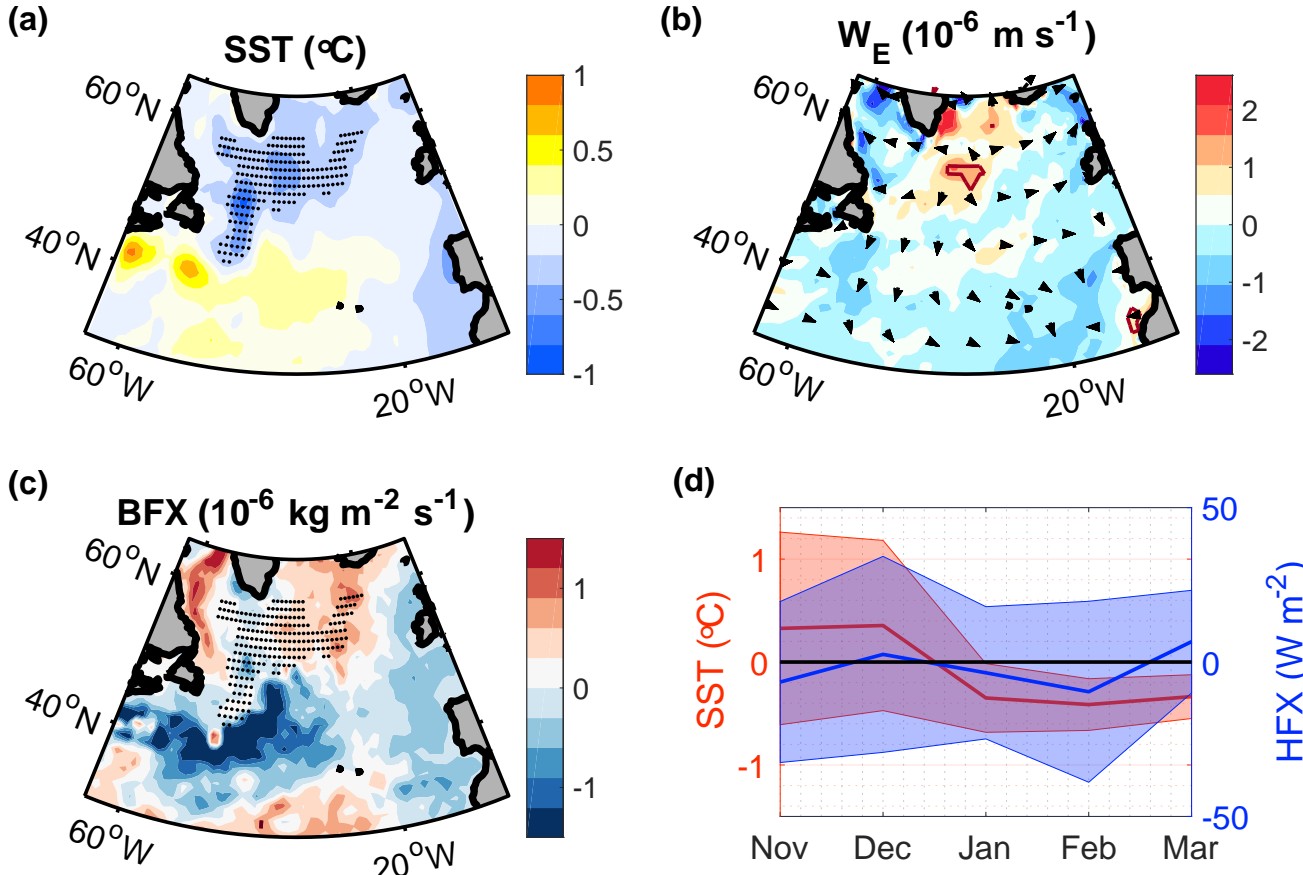

**Figure A5.** (a,b,c) Anomaly of (a) the SST, (b) the vertical Ekman velocity (positive upward) and (c) the buoyancy flux anomaly (positive downward) in the winters (January through March) before the summers of the heat wave composites (Fig. 10). The arrows in (b) indicate the direction of the horizontal Ekman transports and the dots in (a) and (c) mark the region of the mass balance calculations (corresponding to the cold anomaly region). (d) Evolution of the SST and heat flux anomalies during these winters in the cold anomaly region. The shading represents the standard error.



**Figure A6.** Forcing of the 2003 cold anomaly: Anomaly of (a) the SLP in summer 2002 and (b) the absolute dynamic topography, (c) the SST and (d) the sea surface salinity anomaly in winter 2003.

In addition, we investigated the ocean and atmospheric conditions leading to the selected warm summers in 1994, 2003 and 2018 (Fig. 11). The cold anomalies associated with the warm summers in 1994 and 2018 were linked to strong melt-induced freshwater events (Figs. 2-4) while the heat wave in 2003 was linked to a circulation-induced freshwater event (Figs. 8 and 9). Since the associated cold anomaly had an unusually southerly location, we examined it in more detail. We find that the preceding summer was characterised by reduced sea level pressure over Greenland (Fig. A6a), which is representative of a positive NAO and hence supports that the cold anomaly was circulation-induced. However, closer inspection of the absolute dynamic topography reveals that it was associated with a pronounced southward shift of the North Atlantic Current (Fig. A6b), resulting in an enhanced southward expansion of the fresh and cold polar water (Fig. A6c and d).





## Appendix B: Simulated atmospheric response to the freshwater-induced SST in winter

To support the role of the freshwater-induced SST pattern in driving the observed atmospheric response in winters after freshwater events (Fig. 3b), we define an SST index that captures the time variability of the spatial SST pattern linked to freshwater events. Specifically, we map the SST each winter onto the observed SST pattern after freshwater events. The mapping is obtained from a linear least-square fit of the SST each winter to the observed pattern after freshwater events. We then regress the atmospheric stream function at different pressure levels onto this SST index, using 50 SST-forced ensemble simulations from ECHAM5.

**Figure B1.** (a) Variability and (b) distribution of the freshwater-induced SST pattern in winter (January through March), obtained by mapping the SST each winter on the SST after freshwater events (Fig. 2c). $SST_{FW}$ thus represents the temporal variability of the spatial SST pattern linked to freshwater events. (c,d) Regressions of the simulated stream function and winds in winter at (c) 500 hPa and (d) 250 hPa onto the normalised $SST_{FW}$ pattern shown in (a) and (b). The simulations were acquired from 50 SST-forced ensemble members from ECHAM5 over the period 1979–2018. Contours indicate the 95% confidence levels.





The SST-forced model simulations support that the SST pattern associated with freshwater events leads to a significant dipolar atmospheric circulation anomaly over the North Atlantic, extending deep into the troposphere (Fig. B1). Even at 250

hPa, the winds are still following the underlying SST front. Since the simulations were SST-forced, they imply that the dipolar atmospheric circulation pattern is driven by the underlying SST pattern, although the SST pattern can itself also be the result of atmospheric feedbacks. While a detailed description of the involved diagnostics is beyond the scope of this study, the obtained atmospheric response is consistent with theoretical expectations (O'Reilly et al., 2017; Omrani et al., 2019), and the underlying dynamics are well-understood.

**Appendix C: Supplementary figures**

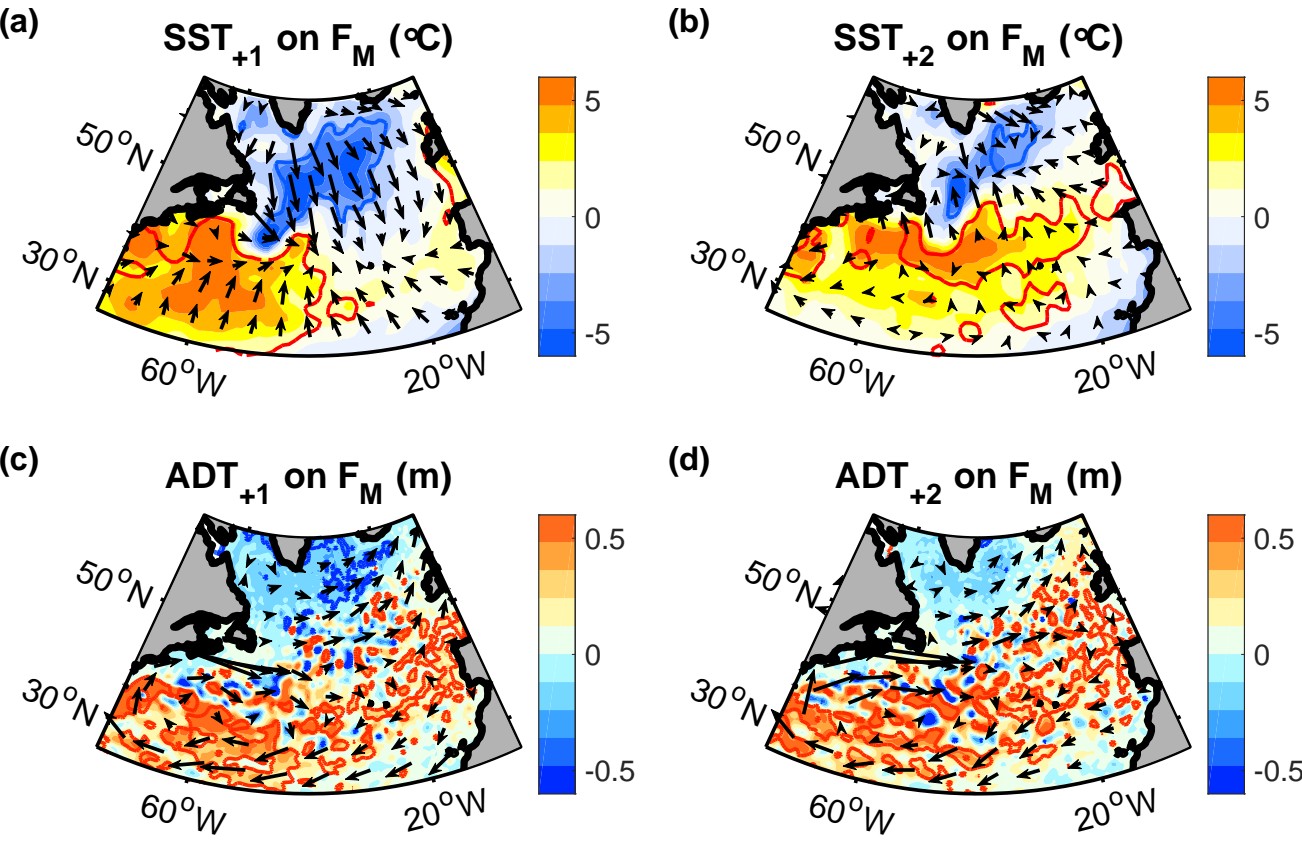

**Figure C1. Detailed flow field after freshwater events.** Regressions of (a,b) the SST and (c,d) the absolute dynamic topography in (a,c) the first winter and (b,d) the second winter after the freshwater events (January through to March) on $F_M$ (Fig. 2b). Contours show the 95% confidence levels. The arrows in (a) and (b) indicate the direction of the wind driven Ekman transports and the arrows in (c) and (d) indicate the direction of the smoothed geostrophic velocity associated with the underlying absolute dynamic topography.



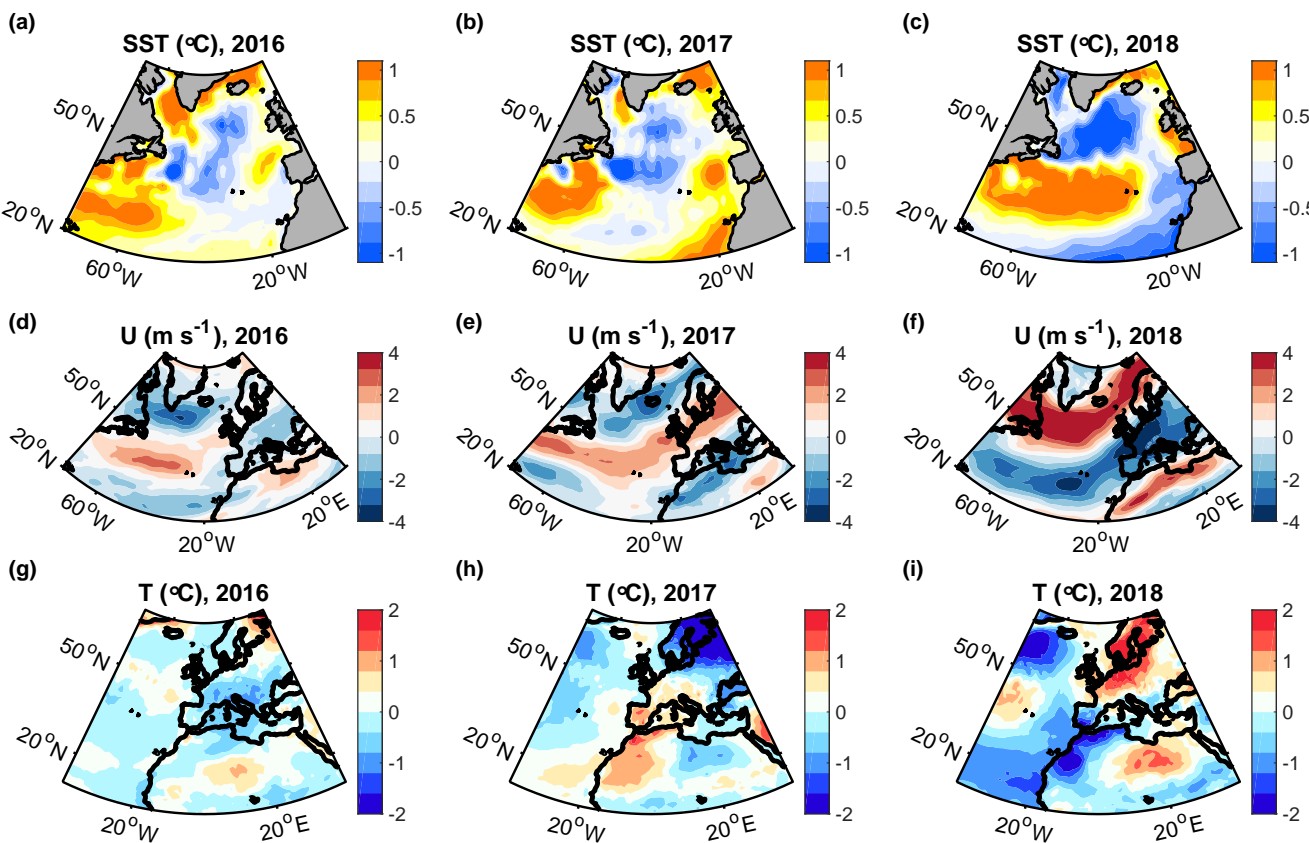

**Figure C2. Evolution of the freshwater events in 2015 and 2016.** Anomalies of (a,b,c) the SST, (c,d,e) the eastward wind component at 700 hPa and (e,f,g) the de-trended 2-m air temperature in the summers (May through to August) of 2016, 2017 and 2018. Shown are the anomalies relative to the climatological summer means.



**Figure C3. Sensitivity to excluding double events.** (a,b,c) Regressions of (a) the SST and 700 hPa winds, (b) the de-trended 2-m air temperature and (c) the accumulated precipitation minus evaporation in the second summer (May through to August) after freshwater events on $F_M$, obtained from the freshwater events in 1980, 1994, 2009, 2012 and 2016. Thus, the first event in all consecutive events has been excluded to avoid overlap. Thick contours indicate the 95% confidence levels. (d) Correlation between $F_M$ and the accumulated precipitation minus evaporation in the 95% confidence region in (c), relative to the climatology mean, with dashed lines representing the standard deviation over all summers.

*Author contributions.* M.O. conceived the study, carried out the analyses and was lead writer of the text. J.S. provided guidance in the model analysis. All authors contributed to the development and writing of the paper.

*Competing interests.* The authors declare that they have no conflict of interest.





*Acknowledgements.* We thank NOAA/OAR/ESRL and the Hadley Centre for providing the SST data, the Copernicus Marine Service for

distributing the altimetry products, the European Centre for Medium-Range Weather Forecasts for developing the reanalysis ERA5 and the

NOAA Physical Sciences Laboratory for facilitating access to the climate model outputs. This study was funded through the grants ACSIS

(NE/N018044/1) and CLASS (NE/R015953/1) from the UK National Environmental Research Council and through the EU Horizon 2020

research and innovation programme Blue-Action (Grant 727852).





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
