# Peer review of "North Atlantic freshwater events influence European weather in subsequent summers"

_Weather and Climate Dynamics, 2021_

## Author Comment (AC1)

**Reviewer 1**

This manuscript examines the impact of summertime NAO events on European weather in subsequent summers. The paper is framed as being the impacts of North Atlantic freshwater events on European weather but the index that is used to depict these "freshwater events" is actually the summer NAO. The paper demonstrates some interesting connections between the summer NAO and the following summer weather. At this point, I'm unsure as to whether this manuscript is acceptable for publication. I have a number of comments on the analysis as outlined below. Overall, I'm giving a recommendation of major revisions to allow the authors to respond to these. My major concerns are that the direct link between the summer NAO and the freshwater events is unclear to me. This may be because I'm not an oceanographer and I haven't read the authors previous papers, so I hope that one of the other reviewers will be able to assess this aspect. I have some other concerns about the statistical methods used and the choices made for the scatter plots as outlined in my comments below.

*We strongly thank the reviewer for reviewing the manuscript and providing many detailed comments and suggestions! The major concern is that the link between the summer NAO, the SST and freshwater is unclear. We will simplify the mass balance analysis, add further explanations, a schematic, and an example to make it easier for the reader to follow the analysis and understand how this link is derived. We will also add a section to motivate the approach of using a mass balance analysis to infer the variability of surface freshwater.*

*Currently available data products for the sea surface salinity have large biases, short time spans or poor spatial and temporal resolutions. For instance, satellite products have biases of up to ∼1 g kg$^{-1}$ in the polar regions and are only available since 2011 (Bao et al., 2019).*

*To overcome these challenges associated with freshwater analyses, we take advantage of the influence of freshwater on the SST in order to infer its variability. Specifically, we select conditions that are associated with pronounced cold anomalies in the subpolar region, and then estimate the extent to which freshwater has contributed these cold anomalies. In the following, we outline the main steps, which will also explained in the main manuscript, rather than only the appendix:*

[Figure]

*Figure 1: Mass budget for a mixed layer of depth h in the cold anomaly region. **A** corresponds to horizontal advection, **M** is an anomalous density flux from beneath the mixed layer, and $\frac{B}{g}$ refers to the density contribution from the surface buoyancy flux.*

*(1) We start with conservation of mass: $\frac{\partial}{\partial t}\int_{-h(t)}^{0}\rho dz = -\frac{B}{g} + M + A$, where B is the buoyancy flux through the surface, g is the gravitational acceleration, $\rho$ is density, M is the mass flux through the base of the mixed layer, and A is horizontal advection (e.g. Griffies and Greatbatch, 2012).*

*(2) Next, we discretise the mass equation and integrate over the winter, using a variable mixed layer depth evolution from $h_0$ to $h_n$:*

$$\rho_n = \frac{h_0}{h_n}\rho_0 + \left(-\frac{B_n}{g} + M_n + A_n\right)\cdot\frac{\Delta t}{h_n}$$

*Here, the subscript $n \in 1..N$ refers to the n'th winter of an arbitrary subset of N winters. Before the winter, the mixed layer ($h_0$) is several tens of metres deep while during the winter, it reaches several hundred metres. Therefore, the density anomaly in the initial shallow mixed layer is distributed over a much larger depth range and the first-term on the right-hand side is negligible compared to the other terms. Any density anomalies beneath the initial, shallow mixed layer are included in $M_n$.*

*(3) We then linearise the equation of state: $\rho_n \approx \rho_m[1 - \alpha(T - T_m) + \beta(S - S_m)]$, where $T$ is the temperature, $S$ is the salinity, and $\alpha$ and $\beta$ are the thermal and haline expansion coefficients. The subscript $m$ refers to an arbitrary reference state, which, for simplicity, is chosen to be the mean over the subset.*

*(4) In order to infer the salinity, we select indices that are well-correlated with a cold anomaly in the subpolar North Atlantic but not with the potential drivers of density anomalies on the right-hand side of the equation. As an educated guess, we start with indices that we expect to be well-correlated with freshwater in the North Atlantic. Thus, we find that the freshwater indices $F_M$ and $F_C$ are associated with pronounced cold anomalies.*

*(5) After evaluating each term in the mass balance for the cold anomaly region, we find that the density increase, resulting from the cold anomaly, is more than one order of magnitude larger than any of the potential drivers on the right-hand side of the equation.*

*This implies that, for the selected indices, there is no anomalous density increase in the cold anomaly region but that the density increase implied by the cold anomaly must be balanced by a density decrease associated with a fresh anomaly $\alpha(T - T_m) \approx \beta(S - S_m)$. The revised manuscript will go through the involved calculations in more detail.*

*To demonstrate that the surface mass balance yields a good approximation of the salinity anomaly, we consider the last two decades, when sufficient Argo float observations are available to test the results. During the strongest observed cold anomalies over this period, which occurred in the winters 2015 and 2016, the correlation between the temperature and salinity anomalies has a p-value of $p = \sim 5.0 \cdot 10^{-242}$, with the regression of salinity on the temperature closely matching the regression predicted by the mass balance analysis (Fig. 2). The approximation, obtained with this method has a root mean square error of $\sim 0.09$ g kg$^{-1}$ and is thus more accurate than any currently available data product for the sea surface salinity. In addition, it provides longer time series, higher resolution, and better coverage than the available in-situ data.*

[Figure]

*Figure 2: Demonstration of the surface mass balance. (a) MLT and (b) MLS are the mixed layer temperature and salinity anomalies during the winters 2015 and 2016, obtained from Argo float profiles (Holte et al., 2017). (c) The red line corresponds to the regression of the mixed layer salinity anomalies on the mixed layer temperature anomalies, while the yellow line corresponds to the approximation obtained from the mass balance analysis. (d) Histogram of the error of the mass balance analysis, corresponding to the difference between the calculated and observed salinity anomalies. The associated root mean square error is ~0.09 g kg⁻¹.*

*Bao, S., Wang, H., Zhang, R., Yan, H., & Chen, J. (2019). Comparison of satellite-derived sea surface salinity products from SMOS, Aquarius, and SMAP. Journal of Geophysical Research: Oceans, 124(3), 1932-1944.*

*Griffies, S. M., & Greatbatch, R. J. (2012). Physical processes that impact the evolution of global mean sea level in ocean climate models. Ocean Modelling, 51, 37-72.*

*Holte, J., Talley, L. D., Gilson, J., & Roemmich, D. (2017). An Argo mixed layer climatology and database. Geophysical Research Letters, 44(11), 5618-5626.*

General comments:

(1) The link to freshwater anomalies and the role of low frequency North Atlantic ocean variability. I am not an oceanographer, so I hope that one of the other reviewers will have the expertise to comment on this. The link between the summer NAO index and the freshwater anomalies was a bit lost on me. My understanding of lines 84-92 is that the authors are assuming that the temperature anomalies associated with the summer NAO are due to freshwater anomalies because they find that the cooling is not strongly related to surface fluxes, wind driven Ekman transports, Ekman pumping and re-emergence of SST anomalies from previous years, so by a process of elimination they conclude that it's freshwater anomalies. But I don't see how the role for other ocean circulation anomalies such as the AMOC or advective heat convergence due to circulation anomalies produced by things other

than the wind driven Ekman transports has been eliminated. The atlantic ocean circulation exhibits variability on long timescales which can be a driver of the NAO and vice-versa (e.g., Zhang et al 2019, Review of Geophysics, 10.1029/2019RG000644 and references therin). It's not clear to me (a) whether it can really be concluded that the SST anomalies are related to freshwater inputs and (b) whether such low frequency variability in the ocean circulation has been appropriately taken into account. The NAO index is being described as a "freshwater index" (l99) but I'm not sure how appropriate this is and I'm not sure that much would be lost by instead referring to it as the NAO index and focussing on the impact of the summertime NAO on the climate in subsequent years.

*We thank the reviewer for pointing out that the derivation of the advective transports and the role of low-frequency variability was unclear. The first question is why we exclude that advective anomalies contribute to the cooling. Away from the boundaries, advective transports must be forced either mechanically through winds or by density gradients. As the reviewer mentions, Ekman transports, driven by the winds, were shown to be negligible.*

*Buoyancy-driven flows, moreover, are the response to existing density gradients in the ocean but cannot create strong gradients on their own (e.g. Wunsch and Ferrari, 2004). In the absence of any external buoyancy forcing, the ocean cannot create density gradients by itself. In the analysis, we show that the surface buoyancy flux is too weak, not significantly correlated with the freshwater indices, and inconsistent to drive the cold anomaly (Figs. A1c and A2c). Thus, it will not create an anomalous buoyancy-driven flow. We conclude that neither wind- nor buoyancy-driven flows can explain the cold anomaly.*

*The second question is whether low-frequency variability has been taken into account. On long timescales, the external freshwater forcing can lead to a reduction in the buoyancy-driven overturning circulation (e.g. Stommel, 1968). In this case, the density decrease associated with the freshening even exceeds the density increase associated with the cold anomaly $\beta\Delta S < \alpha\Delta T$. However, the resulting reduction in the overturning circulation would lead to a positive surface heat flux anomaly in the subpolar region, such that the ocean loses less heat to the atmosphere (e.g. Gulev et al., 2013). The reduction in ocean heat losses can also be understood by considering that freshwater increases the stratification and thus reduces the amount of heat that is available to the atmosphere.*

*As shown in the mass budget, we do not find a significant heat and, in turn, buoyancy flux anomaly (Figs. A1c and d and A2c and d). For instance, the buoyancy flux anomaly associated with $F_M$ events results in a mass decrease of $\sim$7 kg m$^{-2}$, while the cold anomaly implies a mass increase of $\sim$204 kg m$^{-2}$. Thus, the freshwater anomaly implied by the cold anomaly is more than one order of magnitude larger than the freshwater anomaly implied by a potential slowdown of the overturning circulation. In turn, this means that the freshwater increase resulting from a slowdown of the overturning circulation is negligible on the timescales considered.*

*In the revised manuscript, we will clarify that the buoyancy-driven flows are a response, not a driver of density anomalies (Tziperman, 1986; Wunsch and Ferrari, 2004), and we will explicitly state that the freshening, associated with a potential slowdown of the overturning is more than one order of magnitude smaller than the freshening, implied by the cold anomaly. We will further clarify that, on the spatial scales and interannual timescales considered, by far the strongest advective transports of heat and freshwater in the interior subpolar region result from geostrophic flows, both within eddies and as part of the subpolar gyre circulation. However, they do not contribute to the mass budget since geostrophic flows are along density contours.*

Gulev, S. K., Latif, M., Keenlyside, N., Park, W., & Koltermann, K. P. (2013). North Atlantic Ocean control on surface heat flux on multidecadal timescales. Nature, 499(7459), 464-467.

Stommel, H., & Rooth, C. (1968, April). On the interaction of gravitational and dynamic forcing in simple circulation models. In Deep sea research and oceanographic abstracts (Vol. 15, No. 2, pp. 165-170). Elsevier.

Tziperman, E. (1986). On the role of interior mixing and air-sea fluxes in determining the stratification and circulation of the oceans. Journal of Physical Oceanography, 16(4), 680-693.

Wunsch, C., & Ferrari, R. (2004). Vertical mixing, energy, and the general circulation of the oceans. Annu. Rev. Fluid Mech., 36, 281-314.

(2) Detrending: It's stated at line 53-55 that regionally averaged trends were subtracted from the air temperatures to remove the greenhouse gas effects. It doesn't really seem appropriate to me to remove the linear trend from one field but not others. The NAO index that is used clearly has a linear trend in it (Figure 1a). I'd suggest detrending everything or detrending nothing. I'm not arguing that the NAO trend seen in Figure 1 is greenhouse gas forced or that this trend should necessarily be removed, but it just doesn't seem appropriate to me to remove the trend in one field and not in the others. Is the detrending also done on the SSTs? It doesn't make much sense to me to remove the trend from the surface air temperature but not the SSTs.

*We thank the reviewer for making us aware that the removal of the trend was insufficiently explained. Since freshwater has a trend (Fig. 3), trends are part of the dynamic signal we are interested in. For instance, a trend in the freshening would lead to a trend in the cold anomaly, and in turn, a trend in the jet stream shift and so on.*

[Figure]

*Figure 3: Average trend in the sea surface salinity over the last 70 years, inferred from a surface mass balance using Hadley SST data and the reanalysis ERA5.*

*For the air temperature, however, there is an additional, large trend due to increased greenhouse gas concentrations. In contrast to the temperature trend that results from the jet stream shift, the warming trend due to increased greenhouse gases is distributed relatively uniformly. Thus, it can be separated from the warming trend due to the trend in the jet stream shift by averaging the temperature over a sufficiently large area before removing the trend.*

*The removal of the trend in the air temperature is thus based on the assumption that any potential warming trend associated with an SST anomaly over the North Atlantic must be balanced by a cooling trend over the ocean, if the warming and cooling are linked to the same atmospheric instability.*

*We agree with the reviewer that, for consistency, this trend should be removed from all variables. However, we found that neither the SST nor the other atmospheric variables have a significant trend when they are averaged over a large area. For instance, P-E in summer has a trend of $\sim -2.9 \cdot 10^{-4}$ $\pm 3.0 \cdot 10^{-4}$ m year$^{-1}$ when it is averaged over the same area, which is not significant. Likewise, the SST has a trend of -0.0046 ± 0.0062 °C year$^{-1}$ when it is averaged over the North Atlantic (from 0 to 65 °N), which is also not significant. Removing these trends prior to the analysis does not lead to any notable differences in the results.*

*However, upon checking again, we found that the absolute dynamic topography (ADT) also has a significant positive trend when averaged over the North Atlantic. The identified increase is likely due to the long-term ice loss of glaciers, and thermal expansion (Church et al., 2001). In the revised version, we have therefore also removed the ADT trend. This did not appreciably affect the results.*

*In the revised version, we will explain the removal of the trend in more detail in the method section. We will also specify the region used for the averaging and point out that the results are not sensitive to the choice of the region as long as the region is sufficiently large. In addition, we will clarify that none of the other responses (apart from the air temperature and the ADT) has a significant trend when it is averaged. We thank the reviewer for making us check all potential trends again to ensure consistency in the analysis.*

*With regard to the summer NAO, we find that it has a weak trend of 0.01 year$^{-1}$. However, for the reasons stated above, we think that removing this trend would not be meaningful. It is used as an indicator for freshwater. Thus, any manipulation of this time series would affect its representation of the freshwater anomaly and therefore be counterproductive.*

*It is worth noting that the precipitation minus evaporation anomalies and the temperature anomalies show very similar patterns, and that these are consistent with the jet stream shift and, in turn, the cold anomaly. The consistency of the observed patterns across all investigated variables, where only one of them is affected by the direct warming due to increased greenhouse gases (and therefore has a trend when it is averaged), provides compelling evidence that the warming due increased greenhouse gas forcing has successfully been removed.*

*Church, J. A., Gregory, J. M., Huybrechts, P., Kuhn, M., Lambeck, K., Nhuan, M. T., ... & Woodworth, P. L. (2001). Changes in sea level. In , in: JT Houghton, Y. Ding, DJ Griggs, M. Noguer, PJ Van der Linden, X. Dai, K. Maskell, and CA Johnson (eds.): Climate Change 2001: The Scientific Basis: Contribution of Working Group I to the Third Assessment Report of the Intergovernmental Panel (pp. 639-694).*

(3) For the scatter plots, the regions where the correlation is significant at the 95% level is used for the spatial averaging. This seems like cherry picking to me. Of course, the correlations look good because you've chosen them to be that way. It would make more sense to choose a physically motivated region or it would seem to be less cherry picking if a regular spatial region such as a rectangle were chosen. The result is that in Figure 2b there is a correlation of 0.98, which seems quite unbelievable to me, but maybe it isn't if you are just averaging over regions where the correlation is high.

*The significance of the identified relationships is assessed by the thick contours in all regressions that are shown. However, due to the low number of degrees of freedom in the $F_M$ events, the scatter plots are an important addition. This is because the obtained relationships could potentially result from individual outliers or clusters of values. In order to rule out that the significance of the relationships is due to outliers, it is important to show the individual values of the involved timeseries for which the significance is shown.*

*In theory, we could show the scatter plots for all variables but the information would be redundant since we always use the same regressor ($F_M$), and the first and last variables already show that there are no outliers or clusters in the distribution of $F_M$. Instead, the points are evenly distributed. Therefore, it is sufficient to show the values only for the first step in the chain of events. We also show them for the last step (precipitation minus evaporation), to point out the steep regression slopes and thus, the high sensitivity of the P-E response to small variations in the freshwater forcing.*

*Since the scatter plots are needed to confirm the significance of the identified relationships, they must, by definition, include the variables at the locations where a significant link is found. An analysis of the significance outside these regions would not be meaningful.*

(4) Has autocorrelation been accounted for when calculating the significance levels? If not, I think it should be. Clearly each year is not independent and there is some low frequency variability and autocorrelation, as apparent in the NAO index (Figure 2a).

*High auto-correlations across the events can reflect a potential redundancy in the events. Thus, we ensured that the auto-correlations of the freshwater indices are negligible, and we also show them in the appendix. Since we are comparing freshwater events of the same type with each other, rather than different types of freshwater events, we need to consider the autocorrelations across the events for consistency with the analyses. The results are in good agreement with the scatter diagrams which showed that there are not outliers or clusters. To avoid that this information is missed, we will move it to the beginning, where the indices are introduced for the first time, and we will better explain it.*

(5) It is argued that this work reveals new potential to enhance the predictability of European summer weather, but I think for the impacts on European summer weather the results have only been presented in the form of regression coefficients. To make this more relevant for predictability, it might be worth showing the variance explained.

*We thank the reviewer for this excellent suggestion. In the revised manuscript, will make sure to include an additional figure showing the explained variance of European summer weather.*

Comments by line number:

Figure 2: It seems like it would be interesting to have the regression maps for F_C as well as F_M. You use the regions based on the regression onto F_M for both F_M and F_C, so it would be good to see whether the regression map for F_C has a similar spatial pattern to that for F_M or not.

*Thank you for letting us know that the figure raises the question about the corresponding regression maps for $F_C$. In the new version, we will show both regression maps in a single figure. We will also provide the correlation between the two timeseries obtained from each pattern.*

Figure 2 caption: F_M and F_C are only defined in this figure caption. Given their central importance, I think they should also be defined in the text. Furthermore, it would be worthwhile making clear the motivations for this naming convention. It's not very intuitive where the choice of "F_M" and F_C" comes from and I think it would help readers to follow if you make that clear. In the end, I realized that this corresponds to "melt-driven' and "circulation-driven" events and I'm overall just very confused about how this distinction can be made just on the basis of the NAO index, which relates to my general comments above.  I think this needs to be made clearer throughout the manuscript.

*Thank you for pointing out that the motivation for the naming was not clear. We will explain the naming more clearly and have included additional details on the cause of the freshwater anomalies associated with $F_C$ and $F_M$.*

*Freshwater anomalies associated with $F_C$ are characterised by an enhanced offshore advection of fresh, polar water into the subpolar region. We have now shifted the associated figure, showing the circulation anomaly, to the beginning of the analysis, when we introduce the freshwater indices. This circulation anomaly results from the increased windstress curl in the subpolar region during positive NAO years (e.g. Häkkinen et al., 2011).*

*Freshwater anomalies associated with $F_M$, on the other hand, are characterised by more freshwater inside the currents, rather than a change in the currents themselves. This additional freshening is quantified by the negative summer NAO (without sub-sampling) and was derived in Oltmanns et al. following a similar mass balance analysis but for the density change $\frac{\partial \rho}{\partial t}$ from summer to winter rather than absolute anomalies because the focus was on a shallow surface layer of 30 m, allowing to evaluate seasonal differences. The freshening, represented by the negative summer NAO corresponds to the anomalous seasonal freshwater that is added to the subpolar North Atlantic during autumn.*

*The anomalous seasonal freshening associated with a more negative summer NAO applies to all years, without sub-sampling them. However, smaller freshwater anomalies are mixed down before a significant fresh and cold anomaly in winter develops. Therefore, we cannot use the negative summer NAO to obtain absolute anomalies (unless we sub-sample it and only use strong events).*

*When the seasonal surface freshening is too small, the final freshwater anomaly in winter is dominated by changes in the circulation (and thus $F_C$ events), which have the opposite atmospheric driver. This is why we need two freshwater indices. Using both allows to separate the different drivers of fresh, and hence cold, anomalies.*

*The additional seasonal freshening, associated with the negative summer NAO, must be due to runoff, melting or precipitation minus evaporation. After comparing the associated runoff and precipitation anomalies, we find that P-E is anti-correlated with the negative summer NAO, while runoff and melting are correlated (e.g. Hanna et al. 2013). For instance, the correlation between the negative summer NAO and runoff from Greenland and Canada over the last 40 years is r=~0.63 with a p-value of p=~1.5 · 10^{-5}, obtained from the Greenland climate model MAR (Fettweis et al., 2013). However, we do not differentiate between these sources. We only refer to them as $F_M$ events and do not specify whether it is melt from sea ice or glacial ice or runoff, and where the melting originally occurred. We will clarify this in the revised manuscript.*

*So, by (1) showing the change in the ocean circulation, we can link the freshwater anomalies associated with $F_C$ events to a change in the subpolar gyre circulation. By (2) showing the seasonal freshening associated with the negative summer NAO, we can link the freshwater anomalies associated with $F_M$ events to increased seasonal freshwater input into the currents. By (3) comparing the likely roles of*

*runoff and melting with precipitation anomalies, we can link the increased freshwater inside the currents to more runoff and melt from sea ice and glacial ice.*

*We will add a section in the manuscript to derive these links more clearly and thus motivate the naming of the indices. In this section, we will also include further references on the cause of freshwater anomalies, supporting the naming, and the seasonality of freshwater export into the subpolar region during autumn (e.g. Fratantoni and McCartney, 2010; Schmidt and Send, 2007; and references therein). We will also clarify that the additional freshwater during $F_M$ events has multiple origins related to enhanced seasonal runoff, sea ice and glacial melting, but not precipitation.*

*Fettweis, X., Franco, B., Tedesco, M., Van Angelen, J. H., Lenaerts, J. T., van den Broeke, M. R., & Gallée, H. (2013). Estimating the Greenland ice sheet surface mass balance contribution to future sea level rise using the regional atmospheric climate model MAR. The Cryosphere, 7(2), 469-489.*

*Fratantoni, P. S., & McCartney, M. S. (2010). Freshwater export from the Labrador Current to the North Atlantic Current at the Tail of the Grand Banks of Newfoundland. Deep Sea Research Part I: Oceanographic Research Papers, 57(2), 258-283.*

*Häkkinen, S., Rhines, P. B., & Worthen, D. L. (2011). Warm and saline events embedded in the meridional circulation of the northern North Atlantic. Journal of Geophysical Research: Oceans, 116(C3).*

*Hanna, E., Jones, J. M., Cappelen, J., Mernild, S. H., Wood, L., Steffen, K., & Huybrechts, P. (2013). The influence of North Atlantic atmospheric and oceanic forcing effects on 1900–2010 Greenland summer climate and ice melt/runoff. International Journal of Climatology, 33(4), 862-880.*

*Schmidt, S., & Send, U. (2007). Origin and composition of seasonal Labrador Sea freshwater. Journal of Physical Oceanography, 37(6), 1445-1454.*

Figure 3 caption: Maybe explain a bit more what the "absolute dynamic topography" is. Is it just sea surface height?

*Thank you for pointing out that this was unclear. In the revised version, we will clarify that the absolute dynamic topography is the sea level anomaly with respect to the geoid. Absolute dynamic topography thus also allows to show the mean location of the ocean currents, which would be averaged out in the sea level anomalies. However, since we only showed the absolute dynamic topography in regressions, using sea level anomalies would have led to the same results.*

l138: "expansion of the cold anomaly" - perhaps be clear about what this "expansion' is relative to? Is it relative to the previous summer?

*This and the following comments include very helpful suggestions. We thank the reviewer for providing all these suggestions and we will follow all of them to clarify the manuscript.*

Section 4.5: It might be worth making it clear at the beginning here that this is now back to looking at the observations, since in the previous section the focus was on model simulations.

Figure 11 caption: The referencing to the panel labels is messed up in the caption.
Typo's/wording suggestions:
l62: "this index" --> "the NAO index"

l68: suggest "smaller values" --> "more negative values" because the magnitude of the NAO index isn't smaller.

l84: "Fig. 2d" --> "Fig. 2c" (I think d is showing salinity, not temperature)

Figure 4 caption: "The thick contours show the 95% confidence levels" --> "The thick contours encompass regions that are significant at the 95% confidence levels"

l158: "SST-forced" is a bit unclear. Suggest "Simulations performed with prescribed observation-based SSTs".

*We again thank the reviewer for the detailed review, helping us to improve this manuscript!*

---

## Author Comment (AC2)

**Reviewer 2**

The relationship between the Atlantic ocean and the summer climate over Europe is investigated. It is argued that events releasing freshwater into the North Atlantic subpolar gyre are followed by a persistent cooling (warming) over the subpolar (western subtropical) gyre. Such modified SST is linked to warm and dry conditions over western Europe in the next two subsequent summers.

While the overall mechanism seems realistic, the overall presentation of the results is very confusing. I did not understand the link between the SST anomalies analyzed and the freshwater release in the manuscript. Similarly, I was not able to understand many of the analyses presented and the conclusion seems highly speculative. I believe a large amount of work is needed to publish this work in a scientific paper.

*We sincerely thank the reviewer for providing many detailed comments and suggestions! The main concern is that the link between the freshwater and the SST anomalies was unclear. To address this concern, we will include a more detailed derivation, further explanations, a schematic, and an example to make it easier for the reader to follow the analysis and to understand how this link is derived. We will also motivate the approach of using a mass balance analysis more clearly and show, based on a comparison with in-situ observations, that it yields higher accuracies than currently available data products for sea surface salinity over the investigated time period and region.*

*Further details on this, and the clarification of other analyses are provided under the specific comments below.*

**Specific comments**

The Arctic sea ice loss is presented in the introduction and is mentioned in the abstract. But can Arctic sea ice loss release freshwater in summer in the right location? After in the manuscript, L93-98, the Greenland ice sheet melting is mentioned, then the authors say that the scope of the paper is not about understanding the origin of the freshwater. I guess that the introduction and abstract need to be reformulated to have a more balanced picture of the processes releasing freshwater during summer.

*While the origin is not investigated, the cause of the freshening can be inferred from the freshwater indices and associated analyses. We will include a new section in the manuscript that provides more details about the derivation of the causes.*

*In the abstract, we will rephrase the last sentence to "growing freshwater fluxes". In the introduction, we still state that the "loss of sea ice and glacial ice in the Arctic and sub-Arctic regions constitutes a source of freshwater for the North Atlantic". We do not further specify the origin of the freshwater in the introduction.*

In many parts of the manuscript (for instance L59, or legend of Fig. 1), it is argued that a mass balance was used to infer the freshwater release from the SST observation. A reference is given, but can the authors present how this is done. The link between SST and SSS is not obvious and the present paper relies a lot on these previous findings. A presentation of these previous results would improve the manuscript.

*We thank the reviewer for pointing out that the link between the SST and SSS anomalies was not clear.*

*Currently available data products for the sea surface salinity have large biases, short time spans or poor spatial and temporal resolutions. For instance, satellite products have biases of up to ~1 g kg$^{-1}$ in the polar regions and are only available since 2011 (Bao et al., 2019).*

*To overcome these challenges associated with freshwater analyses, we take advantage of the influence of freshwater on the SST in order to infer its variability. Specifically, we select conditions that are associated with pronounced cold anomalies in the subpolar region, and then estimate the extent to which freshwater has contributed these cold anomalies. In the following, we outline the main steps, which will also explained in the main manuscript, rather than only the appendix:*

[Figure]

*Figure 1: Mass budget for a mixed layer of depth h in the cold anomaly region. **A** corresponds to horizontal advection, **M** is an anomalous density flux from beneath the mixed layer, and $\frac{B}{g}$ refers to the density contribution from the surface buoyancy flux.*

*(1) We start with conservation of mass: $\frac{\partial}{\partial t}\int_{-h(t)}^{0}\rho\,dz = -\frac{B}{g} + M + A$, where B is the buoyancy flux through the surface, g is the gravitational acceleration, $\rho$ is density, M is the mass flux through the base of the mixed layer, and A is horizontal advection (e.g. Griffies and Greatbatch, 2012).*

*(2) Next, we discretise the mass equation and integrate over the winter, using a variable mixed layer depth evolution from $h_0$ to $h_n$:*

$$\rho_n = \frac{h_0}{h_n}\rho_0 + \left(-\frac{B_n}{g} + M_n + A_n\right)\cdot\frac{\Delta t}{h_n}$$

*Here, the subscript $n \in 1..N$ refers to the n'th winter of an arbitrary subset of N winters. Before the winter, the mixed layer ($h_0$) is several tens of metres deep while during the winter, it reaches several hundred metres. Therefore, the density anomaly in the initial shallow mixed layer is distributed over a much larger depth range and the first-term on the right-hand side is negligible compared to the other terms. Any density anomalies beneath the initial, shallow mixed layer are included in $M_n$.*

*(3) We then linearise the equation of state: $\rho_n \approx \rho_m[1 - \alpha(T - T_m) + \beta(S - S_m)]$, where $T$ is the temperature, $S$ is the salinity, and $\alpha$ and $\beta$ are the thermal and haline expansion coefficients. The subscript $m$ refers to an arbitrary reference state, which, for simplicity, is chosen to be the mean over the subset.*

*(4) In order to infer the salinity, we select indices that are well-correlated with a cold anomaly in the subpolar North Atlantic but not with the potential drivers of density anomalies on the right-hand side of the equation. As an educated guess, we start with indices that we expect to be well-correlated with*

*freshwater in the North Atlantic. Thus, we find that the freshwater indices $F_M$ and $F_C$ are associated with pronounced cold anomalies.*

*(5) After evaluating each term in the mass balance for the cold anomaly region, we find that the density increase, resulting from the cold anomaly, is more than one order of magnitude larger than any of the potential drivers on the right-hand side of the equation. This implies that, for the selected indices, there is no anomalous density increase in the cold anomaly region but that the density increase implied by the cold anomaly must be balanced by a density decrease associated with a fresh anomaly $\alpha(T - T_m) \approx \beta(S - S_m)$.*

*To demonstrate that the surface mass balance yields a good approximation of the salinity anomaly, we consider the last two decades, when sufficient Argo float observations are available to test the results. During the strongest observed cold anomalies over this period, which occurred in the winters 2015 and 2016, the correlation between the temperature and salinity anomalies has a p-value of $p = \sim 5.0 \cdot 10^{-242}$, with the regression of salinity on the temperature closely matching the regression predicted by the mass balance analysis (Fig. 2). The approximation, obtained with this method has a root mean square error of $\sim 0.09$ g kg$^{-1}$ and is thus more accurate than any currently available data product for the sea surface salinity. In addition, it provides longer time series, higher resolution, and better coverage than the available in-situ data.*

[Figure]

*Figure 2: Demonstration of the surface mass balance. (a) MLT and (b) MLS are the mixed layer temperature and salinity anomalies during the winters 2015 and 2016, obtained from Argo float profiles (Holte et al., 2017). (c) The red line corresponds to the regression of the mixed layer salinity anomalies on the mixed layer temperature anomalies, while the yellow line corresponds to the approximation obtained from the mass balance analysis. (d) Histogram of the error of the mass balance analysis, corresponding to the difference between the calculated and observed salinity anomalies. The associated root mean square error is $\sim 0.09$ g kg$^{-1}$.*

*Bao, S., Wang, H., Zhang, R., Yan, H., & Chen, J. (2019). Comparison of satellite-derived sea surface salinity products from SMOS, Aquarius, and SMAP. Journal of Geophysical Research: Oceans, 124(3), 1932-1944.*

*Griffies, S. M., & Greatbatch, R. J. (2012). Physical processes that impact the evolution of global mean sea level in ocean climate models. Ocean Modelling, 51, 37-72.*

*Holte, J., Talley, L. D., Gilson, J., & Roemmich, D. (2017). An Argo mixed layer climatology and database. Geophysical Research Letters, 44(11), 5618-5626.*

In the interpretation, the authors discuss some sharper SST front between the Gulf Stream and the cold anomaly (L110). The location of the North Atlantic Current is also given by a thick arrow Fig. 3. After, in many parts of the manuscripts (L136-140, or L131) are discussed some shifts of the North Atlantic current. However, the SST anomalies in Fig. 2c show large scale SST anomalies rather than sharp fronts. The North Atlantic current is not well located in Fig. 3. I suggest the authors mention a modification of the SST gradients inducing modification of the lower tropospheric baroclinicity. The investigation of the link with the Gulf stream of North Atlantic current would require showing the mean location of the currents with more accuracy, and I am not sure it is needed to explain the large scale atmospheric response.

*We thank the reviewer for providing all these helpful suggestions.*

*Since the North Atlantic Current carries warm, subtropical water and the subpolar gyre contains cold, polar water, there is a front between the North Atlantic Current and the subpolar gyre, regardless of the exact location of the North Atlantic Current. In any given year, increased cooling of the subpolar region by stronger freshwater events will lead to a stronger meridional SST gradient and in turn, a sharper front between the North Atlantic Current and the subpolar gyre, regardless of where the front is. Therefore, we referred to the sharper SST fronts after stronger freshwater events.*

*The other question concerns the mean location of the North Atlantic Current. Importantly, we referred to the mean over the events. The mean ADT (over the events) is influenced by the mean wind field (over the events) in the same way as the ADT anomalies are influenced by the anomalous wind field in any given year. Considering the coupling with the atmosphere, the mean location of the North Atlantic Current over the events, indicated by the arrow, is already implied by the obtained ADT signal:*

*Suppose the mean location of the North Atlantic Current over the events would occur at a more northern location. In that case, the mean location of the SST front between the subpolar cold anomaly and the North Atlantic Current would also occur further northward. In turn, the region of enhanced baroclinic instability, indicated by the EGR, would be further northward. This would lead to more northward shifted SLP anomalies. The associated Ekman convergence would then also be further north. In turn, the region of increased ADT would be further north, centred over the mean location of the North Atlantic Current, as initially supposed.*

*Thus, the mean location of the North Atlantic Current over the events is implicit in the obtained ADT signal. The anomalous locations of the current can be inferred from the northern and southern bound of the broad band of anticyclonic eddies, representing the anomalous north-south shifts associated with strong and weak events.*

*The revised manuscript includes a figure with the mean ADT and current speed over the events in the appendix to confirm the mean location. However, we agree with the reviewer that the shift in the North*

*Atlantic Current is not needed to explain the large-scale atmospheric feedback. Thus, we may still follow their suggestion and rephrase the text to place less emphasis on the shift of the North Atlantic Current and more emphasis on the overall modification of the SST gradient enhancing the lower tropospheric baroclinicity.*

The manuscript is not based on a quantification of the freshwater released but use the NAO time series from July and August as the starting time series. Why not using the freshwater itself from ERA5? Why not using SSS which could be more related to the freshwater flux. The authors argue that the time series of summer NAO and freshwater are correlated but what does it mean? Can the authors at least suggest some hypothesis behind this statistical relationship? What are the correlations and their p-value? Similarly, the authors used other indices for the freshwater release are used when investigating climate model simulations. The choice of these indices is not well justified, and it seems that different processes are assessed when using different time series, and the link with the freshwater release remains unclear.

*We thank the reviewer for making us aware that the motivation of using indices for the freshwater variability was not clear. ERA5 does not provide sea surface salinity. Moreover, the quality of existing salinity data from ocean reanalyses is strongly limited by the availability of in-situ observations. And even at times and locations when and where re-analyses are constrained by in-situ observations, they still struggle to provide accurate salinity data.*

*For instance, during the freshwater events in 2015 and 2016, when many Argo float profiles were available, the current state-of-the-art ocean reanalysis model ECCO still had biases of over ~0.3 g kg$^{-1}$ compared to Argo float data, despite assimilating these observations (Fig. 3a). The associated root mean square error was ~0.26 g kg$^{-1}$ (Fig. 3b). For the same winters, the root mean square error associated with the surface mass balance analysis was ~0.09 g kg$^{-1}$. In addition, the surface mass balance provides longer timeseries, better spatial coverage and is not sensitive to the availability of in-situ observations as ocean reanalyses.*

[Figure]

*Figure 3: (a) Difference between the sea surface salinity obtained from the ocean reanalysis model ECCO and Argo float observations during the winters 2015 and 2016. Negative values imply that Argo recorded freshwater water than ECCO. (b) Histogram of the sea surface salinity bias.*

*We will include an additional section in the manuscript to demonstrate more clearly how the link between the indices and the freshwater is derived and provide additional justification and motivation of the method.*

The authors chose to subsample their time series so that they have a large relationship between the summer NAO and the SST anomaly in the following winter. In particular, they chose an arbitrary threshold (0.5) and exclude part of the data (one year that seems to be 2014, represented by the yellow point). I do not believe the relationship obtained are representative of the data, as the subsample is somehow selected to have a large relationship. Similarly, later in the manuscript, the time series are again subsampled to build another index in Fig. 8. I am not sure about the interest of doing this.

*We thank the reviewer for pointing out that the motivation for the sub-sampling was unclear. We will clarify the motivation in the manuscript:*

*The objective of this study is to show the relationship between freshwater and its downstream effects, not between the index and the freshwater. The index is used as tool to demonstrate the influences of freshwater. In order to fulfil its purpose, the index must describe the variability of freshwater sufficiently well.*

*Therefore, the dataset is intentionally sub-sampled to obtain a strong relationship between the freshwater and its index. High correlations between the index and the freshwater are a necessity of the method, not a conclusion.*

*Moreover, a high sensitivity of the subpolar cold anomaly to the freshwater index is a necessary requirement for the surface mass balance analysis. If this sensitivity would be too small, the terms on the right-hand side of the mass equation would not be negligible and it would not be possible to infer the variability of freshwater.*

*Since we find that freshwater has two main drivers that are of equal importance, we need two indices. Taking advantage of the near-linear relationships between the freshwater and these two indices (or subsets), we can then use the indices as a tool to demonstrate the influences of freshwater with linear regressions in the subsequent part of the analysis.*

*In the sub-sequent part of the paper, we do not make any assumptions on the relationship between the freshwater and the indices outside the investigated years since the indices are merely used as a tool.*

How the SST impacts the atmosphere and land surfaces in summer is not well discussed or investigated. The impact of the SST on the baroclinic instability and storm tracks are relevant for winter, but in summer other processes might dominate, such as the impacts of the soil moisture or the impact of tropical Atlantic and the intrusion of moist air from the Mediterranean region. In Figs. 4ab only the few wind vectors are shown over the ocean, and it is difficult to see any shift of the jet stream as argued in L136-145. What are the SLP, geopotential height, zonal wind or streamfunction anomalies? Can the Fig. 4ab be extended to include most of Europe and the Mediterranean region? Similarly, when using model results (section 4.4 and Fig. 6), the SLP or the wind is never shown.

*We thank the reviewer for suggesting to include additional diagnostics in the figures. We found increased baroclinic instabilities and zonal wind speeds over the sharper SST gradients in summer, like in winter. The main difference between summer and winter is that, in summer, there is an additional temperature contrast across the coast, likely because the land heats up faster than the ocean. Following the reviewers' suggestion, we will add the zonal velocities over the extended domain (Fig. 4).*

[Figure]

*Figure 4: Regressions of (a,b) the SST with the 700-hPa winds, and (c,d) the zonal velocities on $F_M$ in (a,c) the first and (b,d) the second summer (May through August) after the freshwater events (as in Figure 4 of the manuscript but with the zonal velocities over an extended domain, requested by the reviewer).*

*We will also add arrows in the model analyses. Overall, we find that the simulated winds are similar to the observed winds, albeit with reduced amplitude.*

The authors argue that ''the large-scale dipolar circulation anomaly is reproduced by SST-forced simulations, supporting that it is driven by the ocean (Appendix B)'' L121-122. What do the authors mean by dipolar? Why are the authors present the results in appendix? When looking at the appendix B, another index is used to characterize the freshwater events (why not using the index built on the NAO??), based on SST. Such regression may reflects here the impact of ENSO on the Atlantic ocean, or the impact of tropical Atlantic, and this cannot be interpreted as an impact of the Subpolar Atlantic SST.

*The first question is what we mean by dipolar. We use this term to describe the SLP and stream function anomalies which show a negative anomaly over the subpolar region and a positive anomaly over the subtropical region. To avoid potential confusion, we will remove the term dipolar. Instead, we will specifically refer to "positive and negative circulation anomalies".*

*The second question is why we show the results in the appendix. This is because we already show the observations in the main manuscript and the model results show essentially the same. To reduce the overall number of figures in the manuscript and enhance clarity, we considered it more appropriate to include them in the appendix. Also, we place a stronger emphasis on the summer weather since this is the focus of the study. Thus, for the summer responses, we also show the model results in the main manuscript.*

*The third question is why we use an SST-based index instead of the NAO index. We do this to specifically to demonstrate the influence of the SST on the atmosphere. Showing the link between a model-based NAO index in summer and the atmospheric circulation in the subsequent winter would not be meaningful since freshwater is not well captured by models (e.g. Mecking et al., 2017).*

*We do not attribute the atmospheric circulation anomaly to the cold anomaly alone since it concurs with the subtropical warm anomaly, due to the coupling through the atmosphere. In particular, the wind-driven Ekman feedback, resulting from the atmospheric circulation anomaly leads to upwelling and downwelling in the eastern and western subtropical region, as described by earlier studies. This is consistent with the conclusions drawn and therefore, we refer to "North Atlantic SST" in the text, not just the subpolar SST. We also show the SST pattern along with the index in Figure B1. However, we do not find any significant links between the $SST_{FW}$ index and the SST in the South Atlantic or with ENSO.*

*The SST index we use is based on the region north of 30 degrees and thus, it only covers the North Atlantic. The results are not sensitive to the selected region as long as it covers the increased SST gradient between the subtropical and the subpolar SST anomaly. We thank the reviewer for pointing out that this information was missing in the previous version and will make sure to add it in the revised version.*

*Mecking, J. V., Drijfhout, S. S., Jackson, L. C., & Andrews, M. B. (2017). The effect of model bias on Atlantic freshwater transport and implications for AMOC bi-stability. Tellus A: Dynamic Meteorology and Oceanography, 69(1), 1299910.*

The sea-level anomalies are interesting and show a large band of anti-cyclonic eddies (Fig. 3c). Can the authors discuss these small scale structures? What is the link between the fresh water release and the sea-level anomalies? Maybe a spatial smoothing would be needed to see the large scale structure suggested in the text.

*Thank you for pointing out that the band of anti-cyclonic eddies needs further clarifications. The link between the freshwater release and the anti-cyclonic eddies is not direct but involves feedbacks between the ocean and the atmosphere:*

*(1) A large freshwater anomaly creates a strong cold anomaly, resulting in an increased SST gradient between the cold anomaly and the subtropical water to the south (Fig. 2).*

*(2) The sharper SST gradient, in turn, increases the baroclinic instability, resulting in a more cyclonic circulation north of the SST front and a more anti-cyclonic circulation to the south (Fig. 3a and b).*

*(3) The obtained atmospheric circulation anomaly leads to an enhanced Ekman flow convergence south of the cold anomaly, the region between the subtropical and the subpolar gyre (Fig. 3b).*

*(4) The Ekman flow convergence increases the sea level height and thus modulates the ocean circulation by giving rise to an anti-cyclonic inter-gyre gyre circulation in the region between the subpolar and subtropical gyres (Fig. 3c).*

*The small-scale features arise from the increase in sea level height, which gives rise to instabilities in the ocean with a horizontal scale of the Rossby deformation radius. These are not seen in the SST because we use a 1-degree resolution SST product and a 0.25-degree resolution ADT product. However,*

*if we smooth the ADT to remove the eddies (Fig. 5), it becomes more similar to the SST and shows the shift in the large-scale velocity field more clearly.*

[Figure]

*Figure 5: Regressions of the smoothed absolute dynamic topography in (a) the first winter and (b) the second winter after the freshwater events (January through to March) on $F_M$. Contours show the 95% confidence levels. The arrows in (c) and (d) indicate the direction of the smoothed geostrophic velocity associated with the underlying absolute dynamic topography.*

*In the revised version, we will add further references that provide more details on the individual steps, particularly on the wind-driven ocean gyres and inter-gyre gyre. We will also include a smoothed version of the ADT figure in the appendix. This will help to clarify how the small arrows were obtained.*

The authors should try to reduce the number of figures and appendices, or better summarize their results. I found the appendix not always relevant. For instance appendix A does not help to understand the surface mass balance and the link with the salinity shown in the main manuscript.

*We think that the figures and appendices may be needed to address the reviewers' other comments. Thus, we will add more explanations to clarify the figures and appendices. However, we will carefully re-consider the necessity of each figure to avoid redundancy.*

**Technical details and other comments:**

L35: is the NAO defined as the first or second rotated EOF of monthly 500-hPa geopotential height?

*It corresponds to the first mode. We will add a reference with the detailed derivation of this EOF mode (https://www.cpc.ncep.noaa.gov/data/teledoc/telepatcalc.shtml).*

L40 : why not only use HadISST to avoid discontinuity in the dataset used?

*The NOAA SST has a higher spatial and temporal resolution since it is based on satellite data. If we start the analysis in 1981 and use only the NOAA SST, the results do not change appreciably, suggesting that the discontinuity of the data set does not affect the results. We will further add a data reference to the complete (merged) data product of NOAA and Hadley SST since it is commonly used (https://climatedataguide.ucar.edu/climate-data/merged-hadley-noaaoi-sea-surface-temperature-sea-ice-concentration-hurrell-et-al-2008).*

L47: can the authors specify if u is the module of the wind or the zonal wind?

*We thank the reviewer for pointing out that this was unclear. In the revised version, we will make sure to specify that it is the zonal wind.*

L49-55 : can the author specify the boundary condition used for SST and sea ice, as well as the external forcing (for the two experiments). How are generated the initial conditions?

*We will explain in the data section that the simulations were performed with the pre-scribed, observed SST and sea ice and time varying greenhouse gases (GHG) and ozone. The GHG evolution is based on observed estimates from 1979-2005, and then a RCP6 scenario thereafter. The time varying ozone is based on data from the AC&C/SPARC ozone database. We will also provide the link to the model description, from where the datasets, used for the boundary conditions, can be downloaded (https://psl.noaa.gov/repository/entry/show?entryid=85181601-0435-40be-8461-e282ac884144&output=wiki.view).*

L55:''we subtracted regionally averaged trends from the air temperatures, both in ERA5 and the model output'' -> I do not understand what are mean regionally averaged trends. How are the regions defined? I believe that it is important that all trends be removed before calculating the regression. Are the SST trends removed as well?

*We thank the reviewer for making us aware that the removal of the trend was insufficiently explained. Since freshwater has a trend, trends are an important part of the signal we are interested in. For instance, a trend in the freshening would lead to a trend in the cold anomaly, and in turn, a trend in the jet stream shift and so on.*

*For the air temperature, however, there is an additional, large trend due to increased greenhouse gas concentrations. In contrast to the temperature trend that results from the jet stream shift, the warming trend due to increased greenhouse gases is distributed relatively uniformly. Thus, it can be separated from the warming trend due to the trend in the jet stream shift by averaging the temperature over a sufficiently large area before removing the trend.*

*The removal of the trend in the air temperature is thus based on the assumption that any potential warming trend associated with an SST anomaly over the North Atlantic must be balanced by a cooling trend over the ocean, if the warming and cooling are linked to the same atmospheric instability.*

*We agree with the reviewer that, for consistency, this trend should be removed from all variables. However, we found that neither the SST nor the other atmospheric variables have a significant trend when they are averaged over a large area. For instance, P-E in summer has a trend of $\sim -2.9 \cdot 10^{-4} \pm 3.0 \cdot 10^{-4}$ m year$^{-1}$ when it is averaged over the same area, which is not significant. Likewise, the SST has a trend of -0.0046 ± 0.0062 °C year$^{-1}$ when it is averaged over the North Atlantic (from 0 to 65 °N), which is also not significant. Removing these trends prior to the analysis does not lead to any notable differences in the results.*

*However, upon checking again, we found that the absolute dynamic topography (ADT) also has a significant positive trend when averaged over the North Atlantic. The identified increase is likely due to the long-term ice loss of glaciers, and thermal expansion (Church et al., 2001). In the revised version, we have therefore also removed the ADT trend. This did not appreciably affect the results.*

*In the revised version, we will explain the removal of the trend in more detail in the method section. We will also specify the region used for the averaging and point out that the results are not sensitive to the choice of the region as long as the region is sufficiently large. In addition, we will clarify that none of the other responses (apart from the air temperature and the ADT) has a significant trend when it is averaged. We thank the reviewer for making us check all potential trends again to ensure consistency in the analysis.*

*It is worth noting that the precipitation minus evaporation anomalies and the temperature anomalies show very similar patterns, and that these are consistent with the jet stream shift and, in turn, the cold anomaly. The consistency of the observed patterns across all investigated variables, where only one of them is affected by the direct warming associated with increased greenhouse gases (and therefore has a trend when it is averaged over a large area), provides compelling evidence that the warming due increased greenhouse gas forcing has successfully been removed.*

*Church, J. A., Gregory, J. M., Huybrechts, P., Kuhn, M., Lambeck, K., Nhuan, M. T., ... & Woodworth, P. L. (2001). Changes in sea level. In , in: JT Houghton, Y. Ding, DJ Griggs, M. Noguer, PJ Van der Linden, X. Dai, K. Maskell, and CA Johnson (eds.): Climate Change 2001: The Scientific Basis: Contribution of Working Group I to the Third Assessment Report of the Intergovernmental Panel (pp. 639-694).*

Figure 1 : The regressions shown are regressions of the SST and SSS variation from summer to winter onto the summer NAO. Are the variations calculated from the previous winter (n-1) to summer n? Or from summer n to next winter n? I do not understand why the authors investigate the SST and SSS variations and not the actual SST and SSS anomalies.

*Thank you for pointing out that this was not clear. The variations are calculated from summer (n) to the next winter. Thus, the obtained freshwater anomaly represents the seasonal freshwater that is added to the subpolar region during autumn.*

*Runoff and melting have a pronounced seasonal signal, leading to enhanced freshening of the subpolar region in autumn. By using summer-to-winter differences in the SSS, we can deduce that a more negative summer NAO leads to an amplification of this seasonal freshening in autumn.*

*The anomalous seasonal freshening associated with a more negative summer NAO applies to all years, without sub-sampling them. However, smaller freshwater anomalies are mixed down before a significant fresh and cold anomaly in winter develops. Therefore, we cannot use the negative summer NAO to obtain absolute anomalies (unless we sub-sample it and restrict the analysis to strong events).*

*When the seasonal surface freshening is too small, the final freshwater anomaly in winter is dominated by changes in the circulation (and thus $F_C$ events), which have the opposite atmospheric driver. This is why we need two freshwater indices. Using both allows to separate the different drivers of fresh, and hence cold, anomalies.*

*Thank you for pointing out that this was not clear. We will add a new section in the revised manuscript where we go through this analysis in more detail.*

Figure 1 : I do not understand what are the SSS results? Are they from SSS observations? Can the authors provide more details on the method used to retrieve the SSS?

*The additional seasonal freshening associated with a more negative summer NAO was obtained from a similar mass balance analysis as the freshwater anomalies associated with $F_M$ and $F_C$. However, the mass balance was carried out for a shallow surface layer of fixed depth (30 m), allowing to calculate seasonal differences. Also, it was calculated for all values of the negative summer NAO, without sub-sampling them.*

*Enhanced seasonal freshening always leads to an enhanced cooling rate but not always to absolute cold anomalies. It only leads to absolute cold anomalies, when it exceeds a threshold. Below this threshold the seasonal freshwater is increasingly mixed down and the absolute anomalies are determined by other factors, particularly changes in the circulation. We will clarify this with more details in the approach.*

*We thank the reviewer for pointing out that this was unclear since it is important to interpret the results and understand why we need two freshwater indices.*

Figure 2 : I believe Fig. 2b shows the regression and not the correlation.

*We will be more precise and specify "regression, with r representing the correlation coefficient."*

L66 : The authors find also warming in the western Atlantic at 30°N. Can the authors explain the link between the freshwater flux in the subpolar gyre and the SST anomalies in the subtropical region? It seems that the atmosphere is forcing a large part of the signal, with the so-called tripole pattern as a response to the NAO (Czaja and Frankignoul, 2002).

*Yes, the SST signal is part of atmosphere-ocean coupling. We thank the reviewer for pointing out that this was not clear. In the revised version, we will clarify that the subtropical warming is the expected result of the wind-driven currents associated with the obtained large-scale atmospheric field (Pedlosky, 1996; Vallis, 2017).*

*In particular, the wind field leads to a convergent Ekman transport in the inter-gyre region, and thus increased sea level anomalies. The geostrophic flow associated with the increased sea level results in an anticyclonic inter-gyre gyre circulation, implying a northward shift of the North Atlantic Current, which in turn gives rise to a warm anomaly (Marshall et al. 2001).*

*The currents induced by the geostrophic anomalies are similar for both the first and second winter after the events (Fig. 5, above) but in the first winter, the warm anomaly is only significant in the western regression. We explain this by the southward Ekman transport of fresh and cold polar water on the eastern side, giving rise to enhanced mixing and will clarify this in the revised version.*

*Marshall, J., Johnson, H., & Goodman, J. (2001). A study of the interaction of the North Atlantic Oscillation with ocean circulation. Journal of Climate, 14(7), 1399-1421.*

*Pedlosky, J. (1996). Ocean circulation theory. Springer Science & Business Media.*

*Vallis, G. K. (2017). Atmospheric and oceanic fluid dynamics. Cambridge University Press.*

L64: ''the relationship between the negative summer NAO and the seasonal surface freshening is approximately linear'' -> Can the authors explain how this was assessed and analyzed in the data?

*The relationship was assessed and quantified with the linear regressions. We emphasise the linearity because the regressions were obtained for the full range of the summer NAO, without sub-sampling the data. This is important for understanding the influence of the summer NAO on the seasonal freshening and in turn, for understanding that $F_M$ events are associated with more freshwater inside the current system (rather than a change in the currents). We will clarify this in the manuscript.*

L74 : I think that directional t-tests are not justified here. Do the authors mean 'one-tailed test'? The sign of the regression of the variables studied is not obvious and only two-tailed test are needed here. Can the authors explain?

*This sentence was indeed misleading since we only show regressions. In the figures, we show the 95% confidence interval of the regressions, which are always two-sided in that they have an upper and a lower bound. Whenever we show the correlation coefficients, we will also show the p-value. All of the p-values are relatively small (on the order of 0.001 or smaller). Therefore, it does not matter which significance test is used. We will remove the sentence to avoid confusion.*

L86-87 : ''After evaluating the surface fluxes, wind-driven Ekman transports, Ekman pumping and re-emergence of SST anomalies'' Can the authors explain better where and how these processes are evaluated?

*Each process is evaluated for the cold anomaly region (shown in Figs. A1 and A2). Thus, the terms are integrated over this region and over the winter. Ekman transports, Ekman pumping and surface fluxes are evaluated with the atmospheric reanalysis ERA5. Moreover, in the absence of mechanically-forced upwelling by winds, and an anomalous density flux from the surface forcing, denser water from below the mixed layer cannot move upward, due to gravity. Only water of the same density can be entrained in the mixed layer. In turn, this means that any anomalous cooling from below must be balanced by an anomalous freshening.*

*Subsurface density anomalies can still passively influence the surface density by determining the volume of entrained water and thus, the mixed layer depth, which modulates the influence that the surface buoyancy fluxes have on the surface density. To rule out the possibility that differences in the mixed layer depth have substantial feedback on the surface density, we considered three cases:*

*1) The mixed layer depth in winter is uncorrelated with the freshwater indices: In that case, differences in the mixed layer depth will not significantly influence the regressions and we can approximate the mixed layer depth with the mean mixed layer depth in winter $h_m$. Using an average mixed layer depth, obtained from Argo floats (Holte et al., 2017), we found that the terms on the right-hand side of the mass equation are about two orders of magnitude smaller than those on the left-hand side.*

*2) The actual mixed layer depth is positively correlated with the freshwater indices: In that case, the deeper mixed layers would imply that the terms on the right-hand side of the mass equation become even more negligible since the density anomalies on the left-hand side are multiplied by the mixed layer depth.*

*3) The mixed layer depth is negatively correlated with the freshwater indices: In order to justify having shallower mixed layers, the surface density anomalies must be negative for increased freshwater*

*indices. In turn, this means that the density anomaly associated with the freshening would even exceed the density increase associated with the cold anomaly: $\beta \Delta S < \alpha \Delta T$. Thus, there is no need to evaluate the mass equation further. In this case, the reduced surface density would imply a positive surface heat flux anomaly in the cold anomaly region, such that the ocean loses less heat to the atmosphere. Since the ocean in the subpolar region in winter is always warmer than the air, this reduction in ocean heat losses can be understood by considering that freshwater increases the stratification and thereby reduces the amount of heat that is available to drive the atmosphere.*

*As shown in the mass budget for the cold anomaly region, we do not find a significant heat and, in turn, buoyancy flux anomaly. For instance, the buoyancy flux anomaly associated with $F_M$ events implies a mass decrease of $\sim$7 kg m$^{-2}$, while the cold anomaly implies a mass increase of $\sim$204 kg m$^{-2}$. This implies that any potential over-compensation is negligible on the timescales considered, consistent with in-situ observations from Argo floats and earlier studies (e.g. Zou et al. 2020).*

*In the revised version, we will go through each step of the derivation in detail.*

*Holte, J., Talley, L. D., Gilson, J., & Roemmich, D. (2017). An Argo mixed layer climatology and database. Geophysical Research Letters, 44(11), 5618-5626.*

*Zou, S., Lozier, M. S., Li, F., Abernathey, R., & Jackson, L. (2020). Density-compensated overturning in the Labrador Sea. Nature Geoscience, 13(2), 121-126.*

L89-92: I believe this needs to be better explained. Does the authors assume a perfect density compensation to deduce the SSS? It does not explain how the entrainment below the mixed layer and the re-emergence are evaluated then.

*We do not assume perfect density compensation. We derive it. If the density decrease associated with the freshening would exceed the density increase associated with the cold anomaly ($\beta \Delta S < \alpha \Delta T$), it would lead to a positive surface heat flux anomaly in the cold anomaly region, such that the ocean loses less heat to the atmosphere. However, we find that the heat and buoyancy flux anomalies are too small to allow for a notable underestimation of the freshwater anomaly.*

*Entrainment due to upwelling is evaluated from the surface winds. When averaged over the cold anomaly region, the vertical Ekman velocity amounts to $\sim -1.8 \cdot 10^{-7}$ m s$^{-1}$. Since it is negative, there is enhanced downwelling, rather than upwelling. Multiplied by a typical vertical density gradient of $5.0 \cdot 10^{-4}$ kg m$^{-4}$ across the pycnocline (Holte et al., 2017) and integrated over the winter, the resulting change in the surface density is more than 3 orders of magnitude smaller than the density change resulting from the cold anomaly.*

*Moreover, in the absence of mechanically forced upwelling, and anomalous surface buoyancy fluxes, denser water from below the mixed layer cannot be entrained, due to gravity. The mixed layer can only entrain water of the same density as the surface density. Thus, if anomalously cold water is entrained from below and contributes to the observed cold anomaly, it must also be anomalously fresh. In addition, we ruled out a potential indirect effect of subsurface density anomalies that can arise through their influence on the mixed layer depth (please see our response to your previous comment for further details).*

*Thank you for pointing out that this needed to be better explained. In the revised version, we will go through the derivation in detail.*

L95 : I do not see a pronounced seasonality of the [...] surface freshening in Fig. 1d. Can the authors explain what is the seasonality of the surface freshening and how the anomalies observed reinforce this seasonality?

*In the subpolar region, the seasonal cycle of runoff and melting is very pronounced, with freshwater arriving in the subpolar region during autumn. Seasonal freshwater variations are driven by the annual cycle of river discharge and ice-melt (e.g. Fratantoni and McCartney, 2010; Schmidt and Send, 2007; and references therein).*

*The anomaly shown in Figure 1d represented the additional seasonal freshwater in autumn associated with a more negative summer NAO. Thus, it amplifies the seasonal signal because a more negative summer NAO leads to an enhanced freshening during the subsequent autumn.*

*We point out that the exact distribution of the cold anomalies in winter is not only a function of the freshwater pathways but also results from the surface heat fluxes, which can mix freshwater down. Since the surface heat fluxes are larger in the western subpolar gyre (Fig. A1d), the mixed layers are typically deeper in this region and much of the freshwater is mixed down. Thus, the strongest fresh (and cold) anomalies are found in the eastern subpolar gyre, where the surface fluxes are weaker (Fig. A1d).*

*Fratantoni, P. S., & McCartney, M. S. (2010). Freshwater export from the Labrador Current to the North Atlantic Current at the Tail of the Grand Banks of Newfoundland. Deep Sea Research Part I: Oceanographic Research Papers, 57(2), 258-283.*

*Schmidt, S., & Send, U. (2007). Origin and composition of seasonal Labrador Sea freshwater. Journal of Physical Oceanography, 37(6), 1445-1454.*

L93-98 : I would rather link the freshening with P-E, and I do not understand well the hypothesis that runoff from Greenland is dominant here.

*The additional seasonal freshening, associated with the negative summer NAO, can only be due to runoff, melting or precipitation minus evaporation. After comparing the likely roles of runoff and precipitation anomalies, we find that P-E is anti-correlated with the negative summer NAO, while runoff and melting are correlated, consistent with previous studies (e.g. Hanna et al. 2013). For instance, the correlation between the negative summer NAO and runoff from Greenland and Canada over the last 40 years is r=~0.63 with a p-value of p=~1.5 · $10^{-5}$, obtained from the Greenland climate model MAR (Fettweis et al., 2013). This is in good agreement with the earlier studies on the effect of seasonal runoff and melting on Labrador Sea freshening (Fratantoni and McCartney, 2010; Schmidt and Send, 2007).*

*Precipitation minus evaporation in winter is part of the buoyancy fluxes, evaluated in the surface mass balance. Thus, it is considered in the analysis. However, there is no significant density flux associated with precipitation minus evaporation after freshwater events. When averaged over the cold anomaly region and integrated over the winter, the associated density flux (calculated from ERA5) is ~-0.68 kg $m^{-2}$ which is about three orders of magnitude smaller than the density change resulting from the cold anomaly.*

*We will add a section to provide more details on the cause of the freshening. In it, we will also explain that P-E is in summer is anti-correlated with the negative summer NAO, and that P-E in winter is too small to account for the freshwater anomaly. It also has an inconsistent distribution.*

*Fettweis, X., Franco, B., Tedesco, M., Van Angelen, J. H., Lenaerts, J. T., van den Broeke, M. R., & Gallée, H. (2013). Estimating the Greenland ice sheet surface mass balance contribution to future sea level rise using the regional atmospheric climate model MAR. The Cryosphere, 7(2), 469-489.*

*Hanna, E., Jones, J. M., Cappelen, J., Mernild, S. H., Wood, L., Steffen, K., & Huybrechts, P. (2013). The influence of North Atlantic atmospheric and oceanic forcing effects on 1900–2010 Greenland summer climate and ice melt/runoff. International Journal of Climatology, 33(4), 862-880.*

Figure 3b : The SLP anomalies are huge. Maybe hPa are Pa?

*No, hPa are correct. Please note that these values are regressions, not absolute anomalies. The values are so high because the variations in $F_M$ are very small. They imply that, once the negative summer NAO (and hence the seasonal freshening) exceed a critical threshold, a relatively small further increase is linked to very large atmospheric responses.*

L110: I do not understand what the authors mean with '' after stronger relative to weaker freshwater events ''. What not just say ''after the large freshwater events''?

*Thank you for pointing out that the sentence was confusing. We will simplify it as you suggest.*

Figure 3c: the data used for the ADT need to be presented in the method section.

*We will add further details about the ADT in the method section.*

Figure 3c: the thin black arrow shows the flow implied by the ADT anomaly. What does it mean? I am surprised that such flow is not geostrophic... Can the authors explain how the arrows are computed?

*The arrows were obtained from the smoothed ADT field. Thus, the implied flow is geostrophic. The underlying flow field, resulting from the smoothed ADT, is shown in Figure 5 (page 9 in this document). We will add the smoothed versions of the ADT anomaly in the revised manuscript to clarify that the flow anomalies are geostrophic.*

L123-125 : ''most negative NAO summers are followed by a positive NAO in the subsequent winter'' -> This statement is not supported by the results presented so far, as the regressions shown in observation are built using only 8 winters.

*The most negative NAO summers (for instance in 2015 and 2016) were indeed followed by positive NAO winters. However, to avoid confusion and due to the small sample size, we will remove this sentence.*

L130 : ''the northward shift of the North Atlantic Current is obscured by the southern Ekman flow'' -> Many studies argue that the heat flux is dominant in driving the SST anomalies during the NAO, while the Ekman flow drives weak anomalies (Deser et al., 2010). A more accurate presentation of the terms driving the SST anomalies is required to support this statement.

*We find that the surface heat fluxes associated with the freshwater indices are negligible and inconsistent with the distribution of the SST anomalies (Fig. A1d). The Ekman transports, on the other hand, show a southward flow of the cold, fresh water from the cold anomaly region (Fig. 2c). This flow anomaly is expected to drive enhanced mixing with the warm, subtropical water from the North Atlantic Current to the south. Thus, the Ekman transports can explain why the warm anomaly over the eastern North Atlantic is absent in the first winter, whereas the surface heat fluxes cannot. However, to avoid potential confusion and since this sentence is not needed, we will consider removing it.*

L133-134 : ''an increasingly sharpened SST front all across the eastern boundary of the North Atlantic (Fig. 3d)'' -> Can the authors describe where are these fronts in the eastern Atlantic in Fig. 3d? The SST in the second winter looks similar to that in the first winter, but weaker.

*We will rephrase this sentence and replaced "front" with "warm anomaly". In the second winter, the warm anomaly extends all the way to the east coast of the North Atlantic. In the first winter, it does not (Fig. 2c and 3d). In the revised version, we will show both winters in the same figure, so they are easier to compare.*

L141: ''it shields the regions to the south from the moist air over the Atlantic'' I do not understand this statement. Can the authors reformulate?

*We thank the reviewer for pointing out that the formulation is unclear. In the revised version, we will reformulate this sentence to clarify that the northward turning of the winds over the North Atlantic reduces the advection of moist air masses from the North Atlantic over southern Europe.*

L146: ''the regressions […] are […] characterized by steep slopes and high correlation'' Note that the correlation is never shown in figures, so that the authors may provide hear some number to support this statement. The authors should note that with 8 points, the threshold for a significant correlation is 0.707 for a p-value at 5%. Therefore high correlation does not necessarily mean a significant relationship. I would remove this comment, and I would only comment the level of statistical significance and not the amplitude of the correlation.

*We will rephrase this sentence and only state that the regressions are characterised by steep slopes. Thus, we will follow your suggestion and remove the word "correlation", since we only show the correlation coefficients for the first and the last step of the proposed mechanism.*

Fig. 4ef and Fig. 5ab, the values for P-E are huge. Can the authors check if the map shows correlation and not regression.

*The values for P-E are correct. Please note that these are not absolute anomalies but regressions. They are large, like all the regressions, which is what we mean by "steep slopes of the regressions". There is a high sensitivity to small variations in the freshening, which can be understood by noting that the*

*variations in the underlying F$_M$ index are relatively small. We thank you for making us aware that this was not sufficiently clear. We will clarify this in the revised manuscript.*

Fig. 5cd: I do not see why the authors show these figures… I would remove them.

*We use these figures to better visualise the steep slope of the regressions. In particular, a difference of ~0.3 in F$_M$ (and similarly the summer NAO), corresponds to an additional seasonal freshening of ~0.05 g kg$^{-1}$ averaged over the subpolar region (which is ~0.7 standard deviations). This relatively small additional freshening is followed by a relatively large precipitation decrease of ~15 cm in the subsequent summer, which is a decrease of more than three standard deviations.*

*In conclusion, once the seasonal freshening exceeds a critical threshold (when the negative summer NAO becomes larger than 0.5), a relatively small further increase is linked to very large responses. The figures also show that the high correlation is preserved from the first to the last step of the proposed mechanism. We will add the p-values to the correlation coefficients shown. Since the figures are connected to important statements in the text, we would prefer to keep them.*

L154-157: I do not understand why another index for the freshwater release is used. I do not understand what it is? For instance, L156-157 ''we map the SST each summer onto the observed pattern obtained from freshwater event. The mapping is obtained from a least square fit of the SST […] to the SST pattern obtained from the freshwater events (Figs. 6a and 6b)''. How are the freshwater event defined? Does the author mean that the time series is obtained using a projection onto a spatial pattern over a specified region (that needs to be defined)?

*Yes, the mapping is a projection. The region used is shown by the box in Figure 6b. Since the objective of this analysis is to demonstrate the influence of the SST on the atmosphere, we specifically selected a region and time with a strong SST gradient. For simplicity, we used the SST gradient obtained from the regressions on freshwater events. We will make sure to explain this more clearly in the revised version.*

L159-160 ; ''we find that the observed and simulated atmospheric response agree qualitatively well'' Can the authors explain the difference between Fig. 4 and Fig. 6. It seems that the SST anomaly is different in Fig. 6, with a clear SST tripole, with large impacts on the P-E. It seems that the simulation and models show a feedback between warming over the continent and soil moisture decrease. Did the authors use detrended data? If not, the authors might see here the impacts of global warming over the summer continents.

*Thank you again for noting that the detrending was not sufficiently explained. We did not de-trend the P-E data. The trend in the P-E data is only weak and not significant when it is averaged over a large area. When we remove this trend, the results do not change appreciably. Thus, it seems unlikely that the soil moisture has an influence on the obtained signal. The auto-correlations of P-E are also very small and drop off to ~0 at one year lag. If soil moisture would have a substantial influence, we might expect higher auto-correlations. We thank the reviewer for pointing out that this was not clear and will clarify this in the revised version.*

L173 : ''we […] refer to these freshwater events as circulation-driven events''. The authors argue that explaining the origin of the freshwater anomalies is beyond the scope of the paper. Therefore, I

suggest changing the name of these events, otherwise, the authors should justify how the circulation explains the freshwater anomalies.

*We do not mention the origin of the freshwater but we can infer the likely cause. For circulation-driven freshwater events, we find they are associated with a change in the subpolar gyre circulation (Figure 9a). This signal is exactly what is expected from earlier studies as it implies enhanced offshore advection of fresh and cold polar water into the subpolar gyre (e.g. Häkkinen et al., 2011).*

*We will add a section in the revised manuscript where we explain the causes of the freshwater anomalies associated with $F_M$ and $F_C$ in more detail. In it, we will also clarify the link between the subpolar gyre circulation and interior hydrography and add the associated references.*

*Häkkinen, S., Rhines, P. B., & Worthen, D. L. (2011). Warm and saline events embedded in the meridional circulation of the northern North Atlantic. Journal of Geophysical Research: Oceans, 116(C3).*

L182-183: ''we exclude events, that are preceded by another strong circulation-driven freshwater event, for which F_C is larger than 0.2''. Can the authors justify the choice of 0.2 here? Did they test other values?

*The results of the regressions are not sensitive to this choice. We selected 0.2 to maximise the number of events that are kept, while ensuring that the auto-correlations are negligible at one year lag. We still obtain significant warm and dry anomalies without any further sub-sampling of $F_C$.*

*However, in the revised version, we have revised this approach and specifically sub-sample the $F_C$ years to obtain higher correlations between the index and the freshwater anomalies. This also leads to stronger temperature and precipitation anomalies, similar to the $F_M$ events.*

L184 : ''correlation with the SST gradient'' I do not see any figure of the correlation SST gradient in Fig. 8, but some regression of the SST.

*To avoid confusion, we will remove the figure reference.*

Fig. 10c and 10d :I do not see any level of statistical significance. Does it mean that the SSS or SST in winter is not related to the temperature in summer?

*Thank you for pointing out that the significance lines were not visible. We do find significant SST and SSS anomalies although the significant regions are comparatively small due to the high variability across the events. To demonstrate this high variability across the events, we also show the individual examples in Figure 11.*

Figure 7: how are the outlier defined in the box plot of Figure 7. I believe that showing the 5% and 95% percentile would be useful, as observation seems to be at the edge of the simulated distribution concerning precipitation minus evaportation. The authors should also note that the observation never lies in the interval defined by the first and third quartiles.

*Thank you for suggesting to add 5% and 95% percentiles. We will consider this and also follow your suggestion to stress in the text that the observations never lie in the interval defined by the first and third quartiles. This supports the hypothesis that the simulations may underestimate the influence of the SST signal.*

*We use the standard definition for the maximum possible whisker length, which is 1.5 times the interquartile range. Outliers are any points outside the maximum possible whisker range.*

Figure 10a : How is  T_summer defined? What region? Is it surface air temperature over land only?

*We use the box in Figure 10b to define T_summer. Specifically, we use the average 2-m air temperature over land in this box. However, the results do not change appreciably if we only use the temperature over land or also over the water, as the air temperature over land has a larger variability. Thank you for making us aware that this was unclear. We clarify this in the revised version.*

Figure 11: This figure does not add much to the manuscript.

*The figure helps address many of your earlier comments, for instance regarding the jet stream shift. In addition, it provides examples of different locations and extents of the cold anomalies, which need to be taken into account when interpreting the results from standardised regressions and composites in the preceding analyses. Lastly, the figure helps explain why the regions of the significant cold and fresh anomalies (which will be added in Figure 10) are relatively small. This is because the composite represents an average over events that differ from each other in the exact locations of their cold anomalies. In the revised version, we will clarify these motivations.*

Appendix C: the results are not discussed in the main text. Note sure that the appendix C is needed, the authors may summarize the results by one or two sentence in the main manuscript with ''not shown''.

*Thank you for pointing out that the reason for showing these figures was not clear. In the revised version, we will carefully consider removing them or explain them in more detail if needed.*

References
Deser, C., Alexander, M. A., Xie, S. P., & Phillips, A. S. (2010). Sea surface temperature variability: Patterns and mechanisms. Annual review of marine science, 2, 115-143.
Czaja, A., & Frankignoul, C. (2002). Observed impact of Atlantic SST anomalies on the North Atlantic Oscillation. Journal of Climate, 15(6), 606-623.

*Again, we sincerely thank the reviewer for the many detailed comments, helping us to improve and clarify the manuscript!*

---

## Author Comment (AC3)

**Reviewer 3**

This study investigates a relationship between freshwater anomalies in the North Atlantic and summer European climate up to several years later. The proposed mechanism involves cooling over the subpolar region and warming over the subtropical region that increases the meridional temperature gradient, leading to enhanced baroclinicity that alters the atmospheric circulation. The physical relationships are plausible and there are some interesting implications for predictability. However, I find the approach and manuscript quite confusing, and I believe major revisions would be required before publication.

*We sincerely thank the reviewer for reviewing the manuscript and providing many helpful comments and suggestions! The main concern is that the approach was not clear. To address this concern, we will first motivate the approach more clearly to use a surface mass balance to infer the variability of freshwater. We will also simplify the mass balance analysis and add more explanations to make it easier for the reader to understand how this link is derived. Further details are included under the specific comments below.*

**Main points**

1) I am not clear on whether the analyses actually address the role of freshwater events on European climate. The authors spend quite a bit of time establishing that the relationship between the NAO index and the freshwater events in the period studied is robust and useful (mainly based on previous studies), and hence that the NAO index can be used as a proxy for freshwater anomalies. The justification/explanation comes back in several places throughout the manuscript, perhaps drawing more attention to it than the authors intended. However, I did not fully follow many some aspects of the justification (e.g., a number of other possibilities are eliminated in L86-87, but the explanation is quite brief and as far as I can tell, only focuses on Ekman processes ). The main question I was left with was, why not just use an index of freshwater anomalies? Perhaps there is an obvious answer here, but it didn't come through to me in the manuscript, and makes statements like L102-103 quite unsatisfying.

*We thank the reviewer for pointing out that the motivation of using a surface mass balance to infer freshwater variability was unclear. In the revised version, we will motivate this approach more clearly and go through the derivation in more detail.*

*Currently available data products for the sea surface salinity have large biases, short time spans or poor spatial and temporal resolutions. For instance, satellite products have biases of up to ~1 g kg$^{-1}$ in the polar regions and are only available since 2011 (Bao et al., 2019).*

*To overcome these challenges associated with freshwater analyses, we take advantage of the influence of freshwater on the SST in order to infer its variability. Specifically, we select conditions that are associated with pronounced cold anomalies in the subpolar region, and then estimate the extent to which freshwater has contributed these cold anomalies. In the following, we outline the main steps, which will also explained in the main manuscript, rather than only the appendix:*

[Figure]

*Figure 1: Mass budget for a mixed layer of depth h in the cold anomaly region. **A** corresponds to horizontal advection, **M** is an anomalous density flux from beneath the mixed layer, and $\frac{B}{g}$ refers to the density contribution from the surface buoyancy flux.*

*(1) We start with conservation of mass: $\frac{\partial}{\partial t}\int_{-h(t)}^{0}\rho\,dz = -\frac{B}{g} + M + A$, where B is the buoyancy flux through the surface, g is the gravitational acceleration, $\rho$ is density, M is the mass flux through the base of the mixed layer, and A is horizontal advection (e.g. Griffies and Greatbatch, 2012).*

*(2) Next, we discretise the mass equation and integrate over the winter, using a variable mixed layer depth evolution from $h_0$ to $h_n$:*

$$\rho_n = \frac{h_0}{h_n}\rho_0 + \left(-\frac{B_n}{g} + M_n + A_n\right)\cdot\frac{\Delta t}{h_n}$$

*Here, the subscript $n \in 1..N$ refers to the n'th winter of an arbitrary subset of N winters. Before the winter, the mixed layer ($h_0$) is several tens of metres deep while during the winter, it reaches several hundred metres. Therefore, the density anomaly in the initial shallow mixed layer is distributed over a much larger depth range and the first-term on the right-hand side is negligible compared to the other terms. Any density anomalies beneath the initial, shallow mixed layer are included in $M_n$.*

*(3) We then linearise the equation of state: $\rho_n \approx \rho_m[1 - \alpha(T - T_m) + \beta(S - S_m)]$, where T is the temperature, S is the salinity, and $\alpha$ and $\beta$ are the thermal and haline expansion coefficients. The subscript m refers to an arbitrary reference state, which, for simplicity, is chosen to be the mean over the subset.*

*(4) In order to infer the salinity, we select indices that are well-correlated with a cold anomaly in the subpolar North Atlantic but not with the potential drivers of density anomalies on the right-hand side of the equation. As an educated guess, we start with indices that we expect to be well-correlated with freshwater in the North Atlantic. Thus, we find that the freshwater indices $F_M$ and $F_C$ are associated with pronounced cold anomalies.*

*(5) After evaluating each term in the mass balance for the cold anomaly region, we find that the density increase, resulting from the cold anomaly, is more than one order of magnitude larger than any of the potential drivers on the right-hand side of the equation.*

*This implies that, for the selected indices, there is no anomalous density increase in the cold anomaly region but that the density increase implied by the cold anomaly must be balanced by a density decrease associated with a fresh anomaly $\alpha(T - T_m) \approx \beta(S - S_m)$.*

*The second question is how the terms on the right-hand side of the mass equation were assessed. Buoyancy fluxes are evaluated using ERA5 and were found to be negligible (Fig. A1c and A2c). Moreover, away from the boundaries, advective transports must be forced either mechanically through winds or by density gradients. As the reviewer points out, Ekman transports driven by the winds were shown to be negligible.*

*Buoyancy-driven flows, on the other hand, are the response to existing density gradients in the ocean but cannot create strong gradients on their own (e.g. Wunsch and Ferrari, 2004). In the analysis, we show that the surface buoyancy flux is too weak, not significantly correlated with the freshwater indices, and inconsistent to drive the cold anomaly (Figs. A1c and A2c). Thus, it will not create an anomalous buoyancy-driven flow.*

*On longer timescales, the freshwater forcing can lead to a reduction in the buoyancy-driven overturning circulation (e.g. Stommel, 1968). In this case, the density decrease associated with the freshening would even exceed the density increase associated with the cold anomaly $\beta \Delta S < \alpha \Delta T$. However, the resulting reduction in the overturning circulation would lead to a positive surface heat flux anomaly in the subpolar region, such that the ocean loses less heat to the atmosphere (e.g. Gulev et al., 2013). The reduction in ocean heat losses can be understood by considering that freshwater increases the stratification and thus reduces the amount of heat that is available to the atmosphere.*

*As shown in the mass budget, we do not find a significant heat and, in turn, buoyancy flux anomaly (Figs. A1c and d and A2c and d). For instance, the buoyancy flux anomaly associated with $F_M$ events results in a mass decrease of ~7 kg m$^{-2}$, while the cold anomaly implies a mass increase of ~204 kg m$^{-2}$. Thus, the freshwater anomaly implied by the cold anomaly is more than one order of magnitude larger than the freshwater anomaly implied by a potential slowdown of the overturning circulation. In turn, this means that the freshwater increase resulting from a slowdown of the overturning circulation is negligible on the timescales considered, consistent with Argo float observations and earlier studies (e.g. Zou et al. 2020).*

*We conclude that neither wind- nor buoyancy-driven flows can account for the cold anomaly. On the spatial scales and interannual timescales considered, by far the strongest advective transports of heat and freshwater in the interior subpolar region result from geostrophic flows, both within eddies and as part of the subpolar gyre circulation. However, they do not contribute to the mass budget since geostrophic flows are along density contours.*

*Next, the term $M_n$ is assessed. Therefore, we first evaluated entrainment due to upwelling from the surface winds. When averaged over the cold anomaly region, the vertical Ekman velocity amounts to $\sim -1.8 \cdot 10^{-7}$ m s$^{-1}$. Since it is negative, there is enhanced downwelling, rather than upwelling. Multiplied by a typical vertical density gradient of $5.0 \cdot 10^{-4}$ kg m$^{-4}$ across the pycnocline and integrated over the winter, the resulting change in the surface density is more than 3 orders of magnitude smaller than the density change, resulting from the cold anomaly.*

*Moreover, in the absence of mechanically forced upwelling, and anomalous surface buoyancy fluxes, denser water from below the mixed layer cannot be entrained due to gravity. The mixed layer can only entrain water of the same density as the surface density. Thus, if anomalously cold water is entrained from below and contributes to the observed cold anomaly, it must also be anomalously fresh.*

*Subsurface density anomalies can still passively influence the surface density by determining the volume of entrained water and thus, the mixed layer depth, which modulates the influence that the*

*surface buoyancy fluxes have on the surface density. To rule out the possibility that differences in the mixed layer depth have substantial feedback on the surface density, we considered three cases:*

*1) The mixed layer depth in winter is uncorrelated with the freshwater indices: In that case, differences in the mixed layer depth will not significantly influence the regressions and we can approximate the mixed layer depth with the mean mixed layer depth in winter $h_m$. Using an average mixed layer depth, obtained from Argo floats (Holte et al., 2017), we found that the terms on the right-hand side of the mass equation are about two orders of magnitude smaller than those on the left-hand side.*

*2) The actual mixed layer depth is positively correlated with the freshwater indices: In that case, the deeper mixed layers would imply that the terms on the right-hand side of the mass equation become even more negligible since the density anomalies on the left-hand side are multiplied by the mixed layer depth.*

*3) The mixed layer depth is negatively correlated with the freshwater indices: In order to justify having shallower mixed layers, the surface density anomalies must be negative for increased freshwater indices. In turn, this means that the density anomaly associated with the freshening would even exceed the density increase associated with the cold anomaly: $\beta \Delta S < \alpha \Delta T$. Thus, there is no need to evaluate the mass equation further. Considering that the ocean in the subpolar region in winter is always warmer than the air, an increased stratification would limit the amount of heat available to drive the atmosphere, implying reduced ocean heat (and buoyancy) losses. As shown above, the freshwater anomaly, implied the surface fluxes is about two orders of magnitude smaller than the freshwater anomaly, implied by the cold anomaly. Thus, it is negligible.*

*Lastly, to demonstrate that the surface mass balance yields a good approximation of the salinity anomaly, we consider the last two decades, when sufficient Argo float observations are available to test the results. During the strongest observed cold anomalies over this period, which occurred in the winters 2015 and 2016, the correlation between the temperature and salinity anomalies has a p-value of $p = {\sim}5.0 \cdot 10^{-242}$, with the regression of salinity on the temperature closely matching the regression predicted by the mass balance analysis (Fig. 2). The approximation, obtained with this method has a root mean square error of ${\sim}0.09$ g $kg^{-1}$ and is thus more accurate than any currently available data product for the sea surface salinity. In addition, it provides longer time series, higher resolution, and better coverage than the available in-situ data.*

[Figure]

*Figure 2: Demonstration of the surface mass balance. (a) MLT and (b) MLS are the mixed layer temperature and salinity anomalies during the winters 2015 and 2016, obtained from Argo float profiles (Holte et al., 2017). (c) The red line corresponds to the regression of the mixed layer salinity anomalies on the mixed layer temperature anomalies, while the yellow line corresponds to the approximation obtained from the mass balance analysis. (d) Histogram of the error of the mass balance analysis, corresponding to the difference between the calculated and observed salinity anomalies. The associated root mean square error is ~0.09 g kg$^{-1}$.*

Bao, S., Wang, H., Zhang, R., Yan, H., & Chen, J. (2019). Comparison of satellite-derived sea surface salinity products from SMOS, Aquarius, and SMAP. Journal of Geophysical Research: Oceans, 124(3), 1932-1944.

Griffies, S. M., & Greatbatch, R. J. (2012). Physical processes that impact the evolution of global mean sea level in ocean climate models. Ocean Modelling, 51, 37-72.

Gulev, S. K., Latif, M., Keenlyside, N., Park, W., & Koltermann, K. P. (2013). North Atlantic Ocean control on surface heat flux on multidecadal timescales. Nature, 499(7459), 464-467.

Holte, J., Talley, L. D., Gilson, J., & Roemmich, D. (2017). An Argo mixed layer climatology and database. Geophysical Research Letters, 44(11), 5618-5626.

Stommel, H., & Rooth, C. (1968, April). On the interaction of gravitational and dynamic forcing in simple circulation models. In Deep sea research and oceanographic abstracts (Vol. 15, No. 2, pp. 165-170). Elsevier.

Tziperman, E. (1986). On the role of interior mixing and air-sea fluxes in determining the stratification and circulation of the oceans. Journal of Physical Oceanography, 16(4), 680-693.

*Wunsch, C., & Ferrari, R. (2004). Vertical mixing, energy, and the general circulation of the oceans. Annu. Rev. Fluid Mech., 36, 281-314.*

*Zou, S., Lozier, M. S., Li, F., Abernathey, R., & Jackson, L. (2020). Density-compensated overturning in the Labrador Sea. Nature Geoscience, 13(2), 121-126.*

2) In general, it would be extremely helpful to clarify what this study is about and to choose an analysis strategy that directly addresses the problem. The idea of circulation-induced versus melt-driven freshwater events in section 4.5 came as a surprise to me. In fact, I only realized that F_M and F_C (introduced earlier) are related to this, but had spent quite a bit of the manuscript until then puzzled by the names. Is it really the NAO index that's used to discriminate between these types of events? These ideas should probably be introduced in section 1, as they seem to motivate quite a bit of the study. Interestingly, section 1 as written seems more focused on sea ice loss and the origin of summertime freshwater, but later, the manuscript states that this isn't the focus of the study.

*We thank the reviewer for pointing out that the motivation for using the freshwater indices was unclear. We will clarify the motivation in the manuscript:*

*The objective of this study is to show the relationship between freshwater and its downstream effects, not between the index and the freshwater. The index is used as tool to demonstrate the influences of freshwater. In order to fulfil its purpose, the index must describe the variability of freshwater sufficiently well. Therefore, the summer NAO is intentionally sub-sampled to obtain a strong relationship between the freshwater and its index. Taking advantage of the near-linear relationships between the freshwater and these two indices (or subsets), we can then use the indices as a tool to demonstrate the influences of freshwater with linear regressions in the subsequent part of the analysis.*

*Moreover, a high sensitivity of the subpolar cold anomaly to the freshwater index is a necessary requirement for the surface mass balance analysis. If this sensitivity would be too small, the terms on the right-hand side of the mass equation would not be negligible and it would not be possible to infer the variability of freshwater.*

*We also thank the reviewer for making us aware that the reason for distinguishing between $F_M$ and $F_C$ events was unclear. As the reviewer points out, the origin of the freshwater is not investigated in this study, and this will be clarified in the revised manuscript. However, the cause of the freshening can be inferred from the freshwater indices. Since we find that freshwater has two main drivers that are of equal importance, we need two indices.*

*Freshwater anomalies associated with $F_C$ are characterised by an enhanced offshore advection of fresh, polar water into the subpolar region. We have now shifted the associated figure, showing the circulation anomaly, to the beginning of the analysis, when we introduce the freshwater indices. This circulation anomaly results from the increased windstress curl in the subpolar region during positive NAO years (e.g. Häkkinen et al. 2011).*

*Freshwater anomalies associated with $F_M$, on the other hand, are characterised by more freshwater inside the currents, rather than a change in the currents themselves. This additional freshening is quantified by the negative summer NAO (without sub-sampling) and corresponds to the anomalous seasonal freshwater that is added to the subpolar North Atlantic during autumn. It was derived using a similar mass balance analysis but using seasonal differences, rather than absolute anomalies (Oltmanns et al., 2020).*

*The anomalous seasonal freshening associated with a more negative summer NAO applies to all years, without sub-sampling them. However, smaller freshwater anomalies are mixed down before a significant fresh and cold anomaly in winter develops. Therefore, we cannot use the negative summer NAO to obtain absolute anomalies (unless we sub-sample it and only use strong events).*

*The additional seasonal freshening, associated with the negative summer NAO, must be due to runoff, melting or precipitation minus evaporation. After comparing the associated runoff and precipitation anomalies, we find that P-E is anti-correlated with the negative summer NAO, while runoff and melting are correlated (e.g. Hanna et al. 2013). For instance, the correlation between the negative summer NAO and runoff from Greenland and Canada over the last 40 years is $r=\sim 0.63$ with a p-value of $p=\sim 1.5 \cdot 10^{-5}$, obtained from the Greenland climate model MAR (Fettweis et al., 2013). However, we do not differentiate between these sources. We only refer to them as $F_M$ events and do not specify whether it is melt from sea ice or glacial ice or runoff, or where the melting originally occurred.*

*So, by (1) showing the change in the ocean circulation, we can link the freshwater anomalies associated with $F_C$ events to a change in the subpolar gyre circulation. By (2) showing the seasonal freshening associated with the negative summer NAO, we can link the freshwater anomalies associated with $F_M$ events to increased seasonal freshwater input into the currents. By (3) comparing the likely roles of runoff and melting with precipitation anomalies, we can link the increased freshwater inside the currents to more runoff and melt from sea ice and glacial ice.*

*We will add a section in the manuscript to derive these links more clearly and thus motivate the naming of the indices. In this section, we will also include further references on the cause of freshwater anomalies, supporting the naming, and the seasonality of freshwater export into the subpolar region during autumn (e.g. Fratantoni and McCartney, 2010; Schmidt and Send, 2007; and references therein). We will also clarify that the additional freshwater during $F_M$ events has multiple origins related to enhanced seasonal runoff, sea ice and glacial melting, but not precipitation.*

*Fettweis, X., Franco, B., Tedesco, M., Van Angelen, J. H., Lenaerts, J. T., van den Broeke, M. R., & Gallée, H. (2013). Estimating the Greenland ice sheet surface mass balance contribution to future sea level rise using the regional atmospheric climate model MAR. The Cryosphere, 7(2), 469-489.*

*Fratantoni, P. S., & McCartney, M. S. (2010). Freshwater export from the Labrador Current to the North Atlantic Current at the Tail of the Grand Banks of Newfoundland. Deep Sea Research Part I: Oceanographic Research Papers, 57(2), 258-283.*

*Häkkinen, S., Rhines, P. B., & Worthen, D. L. (2011). Warm and saline events embedded in the meridional circulation of the northern North Atlantic. Journal of Geophysical Research: Oceans, 116(C3).*

*Hanna, E., Jones, J. M., Cappelen, J., Mernild, S. H., Wood, L., Steffen, K., & Huybrechts, P. (2013). The influence of North Atlantic atmospheric and oceanic forcing effects on 1900–2010 Greenland summer climate and ice melt/runoff. International Journal of Climatology, 33(4), 862-880.*

*Schmidt, S., & Send, U. (2007). Origin and composition of seasonal Labrador Sea freshwater. Journal of Physical Oceanography, 37(6), 1445-1454.*

Other points that may or may not be relevant once the main comments are addressed:

3) If the negative NAO index is kept: It's quite confusing to talk about more negative or more positive values of the negative NAO index. I think it's fine to flip the NAO index, but perhaps the text should just talk about higher or lower values of the NAO. Also, I don't think the NAO index was detrended, but 2m temperatures were detrended. What is the reason for this? If trends are kept in, then the autocorrelation needs to be accounted for in subsequent statistical analyses.

*We thank the reviewer for making us aware that the removal of the trend was insufficiently explained. Since freshwater has a trend, trends are part of the dynamic signal we are interested in. For instance, a trend in the freshening would lead to a trend in the cold anomaly, and in turn, a trend in the jet stream shift and so on.*

*For the air temperature, however, there is an additional, large trend due to increased greenhouse gas concentrations. In contrast to the temperature trend that results from the jet stream shift, the warming trend due to increased greenhouse gases is distributed relatively uniformly. Thus, it can be separated from the warming trend due to the trend in the jet stream shift by averaging the temperature over a sufficiently large area before removing the trend.*

*The removal of the trend in the air temperature is thus based on the assumption that any potential warming trend associated with an SST anomaly over the North Atlantic must be balanced by a cooling trend over the ocean, if the warming and cooling are linked to the same atmospheric instability.*

*We agree with the reviewer that, for consistency, this trend should be removed from all variables. However, we found that neither the SST nor the other atmospheric variables have a significant trend when they are averaged over a large area. For instance, P-E in summer has a trend of $\sim -2.9 \cdot 10^{-4}$ $\pm 3.0 \cdot 10^{-4}$ m year$^{-1}$ when it is averaged over the same area, which is not significant. Likewise, the SST has a trend of -0.0046 ± 0.0062 °C year$^{-1}$ when it is averaged over the North Atlantic (from 0 to 65 °N), which is also not significant. Removing these trends prior to the analysis does not lead to any notable differences in the results.*

*However, upon checking again, we found that the absolute dynamic topography (ADT) also has a significant positive trend when averaged over the North Atlantic. The identified increase is likely due to the long-term ice loss of glaciers, and thermal expansion (Church et al., 2001). In the revised version, we have therefore also removed the ADT trend. This did not appreciably affect the results.*

*In the revised version, we will explain the removal of the trend in more detail in the method section. We will also specify the region used for the averaging and point out that the results are not sensitive to the choice of the region as long as the region is sufficiently large. In addition, we will clarify that none of the other responses (apart from the air temperature and the ADT) has a significant trend when it is averaged. We thank the reviewer for making us check all potential trends again to ensure consistency in the analysis.*

*With regard to the summer NAO, we find that it has a weak trend of 0.01 year$^{-1}$. However, for the reasons stated above, we think that removing this trend would not be meaningful. It is used as an indicator for freshwater. Thus, any manipulation of this time series would affect its representation of the freshwater anomaly and therefore be counterproductive.*

*Church, J. A., Gregory, J. M., Huybrechts, P., Kuhn, M., Lambeck, K., Nhuan, M. T., ... & Woodworth, P. L. (2001). Changes in sea level. In , in: JT Houghton, Y. Ding, DJ Griggs, M. Noguer, PJ Van der Linden,*

*X. Dai, K. Maskell, and CA Johnson (eds.): Climate Change 2001: The Scientific Basis: Contribution of Working Group I to the Third Assessment Report of the Intergovernmental Panel (pp. 639-694).*

4) Some of the oceanography concepts could be better explained for the non-oceanographers, and the same goes for the atmospheric concepts. e.g., L89 "the mass increase, implied by the cold anomaly,..."; L112-115 connection between poleward vorticity transport and momentum tranfer from STJ to EDJ, L138-140 is there some relevant theory for the time scales behind the delay in the shift of the North Atlantic Current?

*We thank the reviewer for making us aware that some concepts need to be better explained. In the revised version, we will provide more background information and associated references.*

*Regarding your question on the shift in the North Atlantic Current: We do not find a delay in the shift. The signal in the absolute dynamic topography shows that the shift already occurs in the first winter (Fig. 3 below), as expected from the mechanism (specifically the Ekman convergence in the inter-gyre region, leading to a more anti-cyclonic inter-gyre gyre circulation, Marshall et al., 2001). However, in the first winter after the events, the resulting northward shift of the North Atlantic Current is not seen all the way to the eastern side of the North Atlantic in the SST field, as it is partly covered by the cold anomaly. We will clarify this in the revised manuscript.*

[Figure]

*Figure 3: Regressions of the smoothed absolute dynamic topography in (a) the first winter and (b) the second winter after the freshwater events (January through to March) on $F_M$. Contours show the 95% confidence levels. The arrows in (c) and (d) indicate the direction of the smoothed geostrophic velocity associated with the underlying absolute dynamic topography.*

*Once the forcing of the wind-driven Ekman transports ceases, the inter-gyre circulation anomaly is maintained by geostrophic balance. The decay of the flow anomaly therefore depends on the strength of the initial signal.*

*In the revised version, we will add further references that provide more details on the individual steps, particularly on the wind-driven ocean gyres and inter-gyre gyre circulation. We will also include a smoothed version of the ADT figure in the appendix to help to clarify how the northward shift was derived. In addition, we will show both winters in the same figure, so they are easier to compare.*

5) L 148 "succesfully extracts..." Perhaps related to my general confusion about F_C and F_M, I don't have a good feel for how downstream effects from other drivers and IV would influence F_M, so this statement is difficult to understand.

*The sentence is meant to confirm that the method of first establishing an index with a strong relationship to freshwater, and then using this index to demonstrate the downstream effects of freshwater, was successful. Specifically, it was successful in identifying a coherent mechanism, where all the individual steps follow the chain of events expected from theory and earlier studies.*

*In the revised version, we will rephrase this sentence accordingly. We will also be more cautious in our conclusions and acknowledge the potential existence of other (yet unknown) drivers.*

6) L224: This first line of the conclusions is not representative of the main message of this study, is it?

*We will remove this sentence in the revised version.*

Technical points:

-L61: "well-correlated" should be quantified if the NAO is kept

*Thank you for this suggestion. In the revised version, we will add the corresponding correlation coefficients between the freshwater indices and the freshwater, rather than only the SST.*

-Fig 1a is encapsulated in Fig. 2a - maybe don't need both?

*It may be easier for the reader to interpret Figure 1 and Figure 2 if panel a is kept in both. The colouring may help understand the approach in Figure 2, and it adds a reference for the strength of the regression amplitudes in Figure 1. However, we will carefully consider removing one of the panels in the revised version to remove any redundancy.*

-L84: Fig 2d is SSS?

*Thank you for pointing this out. We will correct the figure reference.*

-L127: the increase in sea level height is just in the subtropical gyre?

*The increase in sea level height and associated circulation anomaly is strongest in the inter-gyre region, between the subtropical and subpolar gyre. Positive and negative anomalies in the sea level height therefore indicated northward and southward shifts of the boundary between the subpolar and the subtropical gyres, and thus the North Atlantic Current (Marshall et al., 2001). We will clarify this in the revised version.*

*Marshall, J., Johnson, H., & Goodman, J. (2001). A study of the interaction of the North Atlantic Oscillation with ocean circulation. Journal of Climate, 14(7), 1399-1421.*

*We again thank the reviewer for all the helpful comments and suggestions, helping us to improve this manuscript!*